# An unexpected noncarpellate epigynous flower from the Jurassic of China

Qiang Fu[1], Jose Bienvenido Diez[2], Mike Pole[3], Manuel García Ávila[2,4], Zhong-Jian Liu[5]*, Hang Chu[6], Yemao Hou[7], Pengfei Yin[7], Guo-Qiang Zhang[5], Kaihe Du[8], Xin Wang[1]*

[1]CAS Key Laboratory of Economic Stratigraphy and Paleogeography, Nanjing Institute of Geology and Palaeontology and Center for Excellence in Life and Paleoenvironment, Chinese Academy of Sciences, Nanjing, China; [2]Departamento de Geociencias, Universidad de Vigo, Vigo, Spain; [3]Queensland Herbarium, Brisbane Botanical Gardens Mt Coot-tha, Toowong, Australia; [4]Facultade de Bioloxía, Asociación Paleontolóxica Galega, Universidade de Vigo, Vigo, Spain; [5]State Forestry Administration Key Laboratory of Orchid Conservation and Utilization at College of Landscape Architecture, Fujian Agriculture and Forestry University, Fuzhou, China; [6]Tianjin Center, China Geological Survey, Tianjin, China; [7]Key Laboratory of Vertebrate Evolution and Human Origin of Chinese Academy of Sciences, Institute of Vertebrate Paleontology and Paleoanthropology and Center for Excellence in Life and Paleoenvironment, Chinese Academy of Sciences, Beijing, China; [8]Jiangsu Key Laboratory for Supramolecular Medicinal Materials and Applications, College of Life Sciences, Nanjing Normal University, Nanjing, China

**Abstract** The origin of angiosperms has been a long-standing botanical debate. The great diversity of angiosperms in the Early Cretaceous makes the Jurassic a promising period in which to anticipate the origins of the angiosperms. Here, based on observations of 264 specimens of 198 individual flowers preserved on 34 slabs in various states and orientations, from the South Xiangshan Formation (Early Jurassic) of China, we describe a fossil flower, *Nanjinganthus dendrostyla* gen. et sp. nov.. The large number of specimens and various preservations allow for an evidence-based reconstruction of the flower. From the evidence of the combination of an invaginated receptacle and ovarian roof, we infer that the seeds of *Nanjinganthus* were completely enclosed. Evidence of an actinomorphic flower with a dendroid style, cup-form receptacle, and angiospermy, is consistent with *Nanjinganthus* being a *bona fide* angiosperm from the Jurassic, an inference that we hope will re-invigorate research into angiosperm origins.
DOI: https://doi.org/10.7554/eLife.38827.001

*For correspondence:
zjliu@fafu.edu.cn (Z-JL);
xinwang@nigpas.ac.cn (XW)

Competing interests: The authors declare that no competing interests exist.

## Introduction

Despite the importance of, the great interest in and intensive effort spent on investigating angiosperms, a controversy remains as to when and how this group came into existence. Since the time of Darwin, some scholars have proposed that angiosperms existed before the Cretaceous (*Smith et al., 2010*; *Clarke et al., 2011*; *Zeng et al., 2014*; *Buggs, 2017*), although the conclusion 'there are no reliable records of angiosperms from pre-Cretaceous rocks' made almost 60 years (*Scott et al., 1960*) seemed to be recently re-confirmed (*Herendeen et al., 2017*). Such uncertainty makes answers to many questions about the phylogeny and systematics of angiosperms tentative. Some reports of early angiosperms (i.e., *Monetianthus* (*Friis et al., 2001*)) are based on a single specimen, which restricts further testing and confirming. Better and more specimens of early age and with

**eLife digest** From oranges to apples, flowering plants produce most of the fruits and vegetables that we can see on display in a supermarket. While we may take little notice of the poppy fields and plum blossoms around us, how flowers came to be has been an intensely debated mystery.

The current understanding, which is mainly based on previously available fossils, is that flowers appeared about 125 million years ago in the Cretaceous, an era during which many insects such as bees also emerged. But not everybody agrees that this is the case. Genetic analyses, for example, suggest that flowering plants are much more ancient. Another intriguing element is that flowers seemed to have arisen during the Cretaceous 'out of nowhere'.

Fossils are essential to help settle the debate but it takes diligence and luck to find something as fragile as a flower preserved in rocks for millions of years. In addition, digging out what could look like a bloom is not enough. It is only if the ovules (the cells that will become seeds when fertilized) of the plant are completely enclosed inside the ovary before pollination that researchers can definitely say that they have found a 'true' flower.

Now, Fu et al. describe over 200 specimens of a new fossil flower that presents this characteristic, as well as other distinctive features such as petals and sepals – the leaf-like parts that protect a flower bud. Called *Nanjinganthus*, the plant dates back to more than 174 million years ago, making it the oldest known record of a 'true' flower by almost 50 million years. Contrary to mainstream belief, this would place the apparition of flowering plants to the Early Jurassic, the period that saw dinosaurs dominating the planet. This discovery may reshape our current understanding of the evolution of flowers.

DOI: https://doi.org/10.7554/eLife.38827.002

features unique to angiosperms are highly sought-after to test related evolutionary hypotheses. Here, we report an unusual actinomorphic flower, *Nanjinganthus* gen. nov., from the Lower Jurassic based on the observations of 264 specimens of 198 individual flowers on 34 slabs preserved in various orientations and states (*Supplementary file 1*). The abundance of specimens allowed us to dissect some of them, thus demonstrate and recognize a cup-form receptacle, ovarian roof, and enclosed ovules/seeds in *Nanjinganthus*. These features are consistent with the inference that *Nanjinganthus* is an angiosperm. The origin of angiosperms has long been an academic 'headache' for many botanists, and we think that *Nanjinganthus* will shed a new light on this subject.

## Results

### Genus

*Nanjinganthus* gen. nov.

### Generic diagnosis

Flowers subtended by bracts. Bracts fused basally. Flowers pedicellate, actinomorphic, epigynous, with inferior ovary. Sepals 4–5, rounded in shape, each with usually 4–6 longitudinal ribs in the center and two lateral rib-free laminar areas, attached to the receptacle rim with their whole bases, surrounding the petals when immature, with epidermal cells with straight cell walls. Petals 4–5, cuneate, concave, each with usually 5–6 longitudinal ribs in the center and two lateral rib-free laminar areas, with rounded tips, surrounding the gynoecium when immature, with epidermal cells with straight cell walls. Gynoecium in the center, unilocular, fully closed by a cup-form receptacle from the bottom as well as sides and by an integral ovarian roof from the above. Style centrally attached on the top of the ovarian roof, dendroid-formed. One to three seeds inside the ovary, elongated oval, hanged on the ovarian wall by a thin funiculus, with the micropyle-like depression almost opposite the chalaza.

### Type species

*Nanjinganthus dendrostyla* gen. et sp. nov.

## Etymology

*Nanjing*- for Nanjing, the city where the specimens were discovered, and -*anthos* for 'flower' in Latin.

## Type locality

Wugui Hill, Sheshan Town, Qixia District, Nanjing, China (N32°08″ 19′ , E118°58″ 20′) (*Figure 1—figure supplement 1*).

## Horizon

The South Xiangshan Formation, the Lower Jurassic.

## Species

*Nanjinganthus dendrostyla* gen. et sp. nov.

## Specific diagnosis

the same as the genus.

## Description

The flowers are frequently concentrated and preserved in groups on certain bedding surfaces (*Figures 1a–g* and *2a–b*), although many of them are preserved as isolated individuals on other slabs.

### Flower bud

A flower bud is preserved as a coalified compression, 6.4 mm long and 3 mm wide, with characteristic longitudinal ribs on the sepals and petals (*Figure 2g*). The sepals are estimated to be 1.3–2.2 mm long and approximately 1.8 mm wide (*Figure 2g*). The petals (including the eclipsed portion) are estimated to be approximately 3.7 mm long (*Figure 2g*). The receptacle/ovary is approximately 3 mm in diameter (*Figure 2g*).

### Mature flower

The flowers are preserved in various states (including coalification), with cup-form receptacle, epigynous with an inferior ovary, 8.4–10.7 mm in length and 6.8–12.8 mm in diameter, actinomorphic in the bottom and top views (*Figures 1a-g*, *2a-f,h*, *3a-b,d-f*, *4a-b,d,g*, *5e-i*, *6a,f,j,l* and *7a,e*). The pedicel is approximately 0.76 mm in diameter (*Figure 6a,b*). Basally fused bracts 0.7–3.7 mm long are observed at the bottom in a few flowers, and a stoma is seen on a bract (*Figures 4g-h*, *7e,h,* and *8h*). The receptacle is cup-form, 3–4.8 mm in diameter and 2–4.5 mm high, surrounded by a 0.3 mm thick wall in the bottom and sides, and covered by an ovarian roof from the above (*Figures 2h–i*, *4d*, *5h*, *6a–b* and *7a,e–f,i*). Scales are attached on the sides of the receptacle/ovary (*Figures 3a–b*, *4a,g–h*, *5i* and *7a,e,i*). The sepals are 1.7–3 mm long and 2.7–4.3 mm wide, with two lateral rib-free laminar areas and usually four longitudinal ribs in the center, and attached to the receptacle rim with their whole bases (*Figures 2c-f*, *3d-f*, *4a,b,d-e*, *5i,l*, *6l*, *7e,i*, *9a*). The elongated epidermal cells are, 44–156 µm x 33–54 µm, with straight cell walls in the middle region, while isodiametric epidermal cells 16–71 µm x 10–54 µm are seen in the lateral laminar areas of the sepals (*Figures 8g*, *9h-k* and *10d-f*). The petals are 3.1–6.6 mm long, 1.9–5.4 mm wide, compressed to only about 11 µm thick, with two lateral rib-free laminar areas, a cuneate base, and 5–6 longitudinal ribs in the center, located inside the sepals on the rim of the receptacle (*Figures 2c–f,h,j*,*4a,b,d*,*6a,f*,*7a,e,i*,*8a–f*,*9a–g*). The ribs are approximately 0.12 mm wide, forking only basally, with elongated epidermal cells with straight cell walls, 32–144 µm x 17–30 µm on the abaxial and 19–72 µm x 13–29 µm on the adaxial (*Figures 8a-f*, *10a-b*). The lateral laminar areas are free of ribs, and each is approximately 1.2 mm wide, with isodiametric epidermal cells 23–64 µm x 18–37 µm (*Figures 8e*, *9g*). A possibly immature stoma is seen on one of the petals (*Figure 10c*). An unknown organ (staminode?) is seen once on the rim of the receptacle (*Figure 6a,m*). The ovarian roof is horizontal, with smooth integral outer and inner surfaces, 0.14–0.22 mm thick, with a style vertically located on its center (*Figures 4c*, *5h*, *6f* and *7a–c,e–g*). The style is 0.3–0.8 mm in diameter, with lateral branches that make the width of the style 3–6 mm (*Figures 2h–i*, *3a–b*, *5i–j*, *6a,c* and *7a–d*). The basalmost pair of the lateral

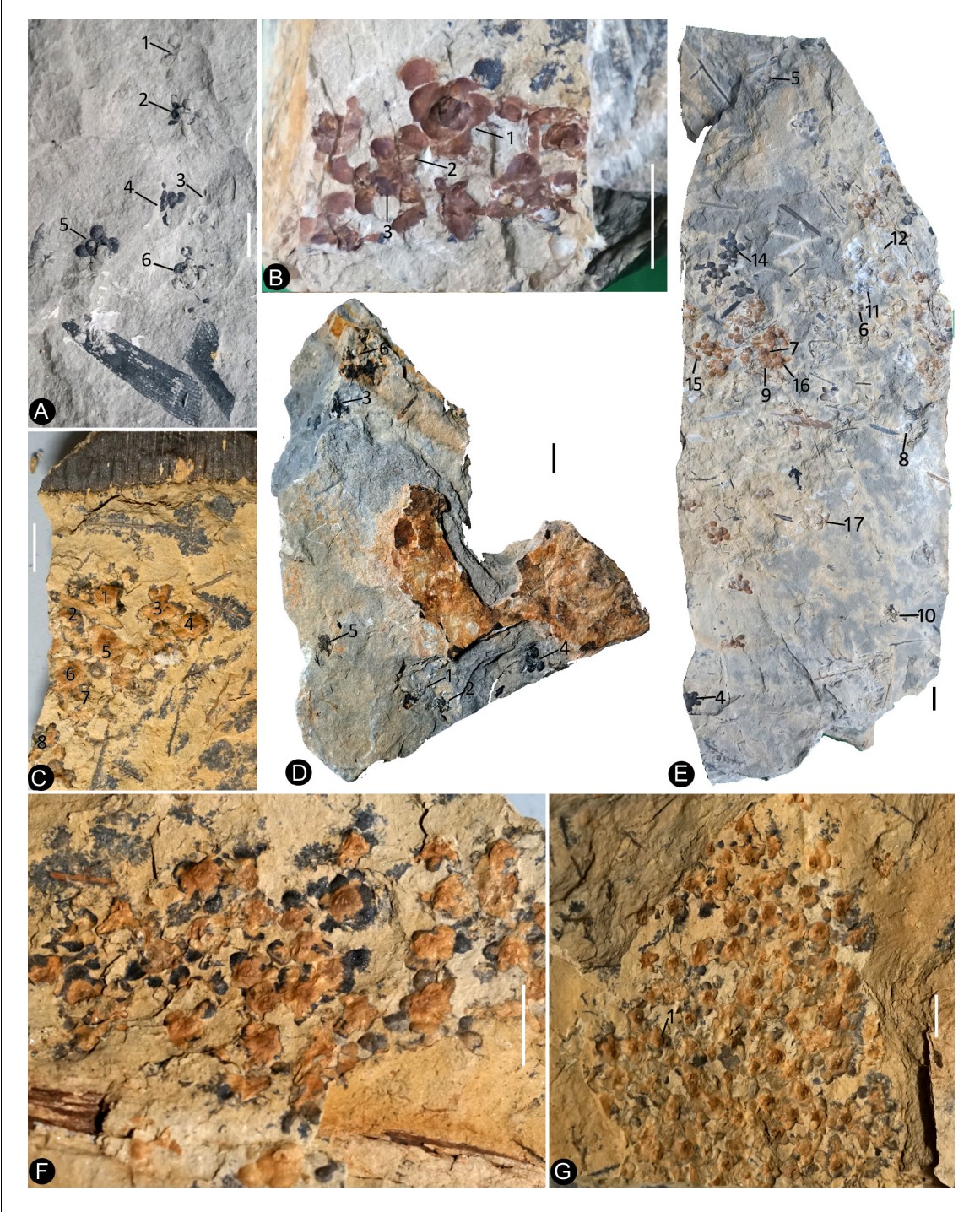

**Figure 1.** Siltstone slabs bearing *Nanjinganthus.* All bars are 1 cm long. (A) Six flowers (1-6) on the same slab, and an associated triangular leaflet with parallel venation. PB22227. (B) Several flowers on the same slab. 1–3 are shown in detail in *Figures 2f* and *6d,e*. PB22226. (C) Several flowers (1-8) on the same slab and the associated *Nilssonia parabrevis* (top). PB22220. (D) Several flowers (1-6) on the same slab. 1–3 are shown in detail in *Figures 2h* and *3a–c*. PB22224. (E) Many flowers on the same slab. Some of the numbered ones are shown in detail in later figures. PB22222a. (F) A slab with numerous flowers. PB22221. (G) A slab almost fully covered with flowers. PB22228.

DOI: https://doi.org/10.7554/eLife.38827.003

The following figure supplement is available for figure 1:

**Figure supplement 1.** The type fossil locality of *Nanjinganthus*, Nanjing in China and isotopic dating.

DOI: https://doi.org/10.7554/eLife.38827.004

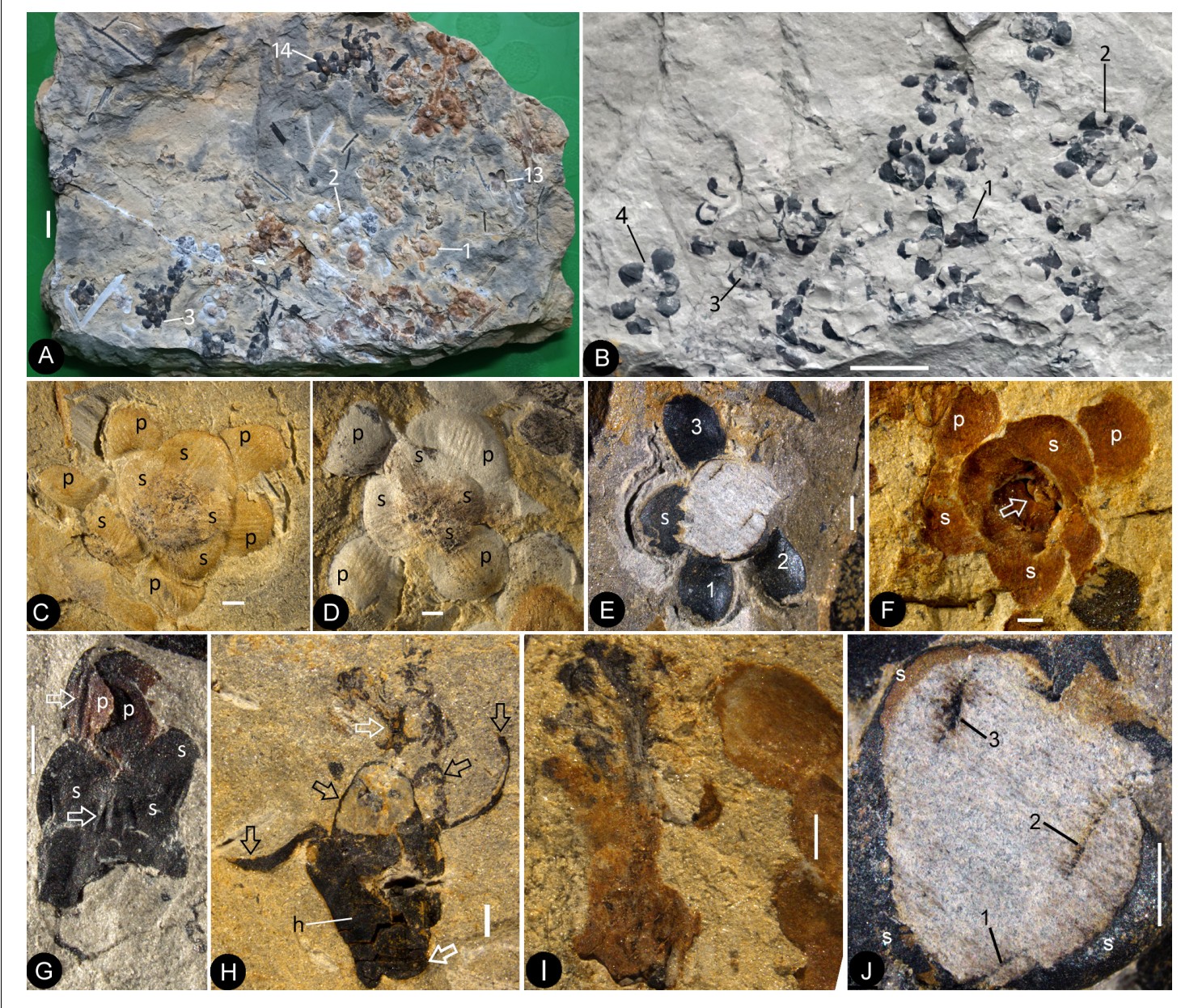

**Figure 2.** Flowers of *Nanjinganthus* preserved in different states and their details. Bar = 1 mm except otherwise annotated. (A) Numerous flowers preserved on a single slab. Some of the numbered ones are detailed in later figures. PB22222B. Bar = 1 cm. (B) Numerous coalified flowers on the same slab. Some of the numbered ones are detailed in *Figure 3d–e*. PB22223. Bar = 1 cm. (C) Bottom view of Flower 1 in *Figure 2a*, showing five sepals (s) and five petals (p) with longitudinal ribs. PB22222B. (D) Bottom view of Flower 2 in *Figure 2a*, showing four sepals (s) and four petals (p) with longitudinal ribs. PB22222B. (E) Bottom view of the flower in *Figure 3f*, showing a sepal (s) and three petals (p) radiating from the center, which is obliquely broken to show the relationship among the sepals and petals as in *Figure 2j*. PB22278. (F) Top view of Flower 1 in *Figure 1b* with sepals (s), petals (p), and seeds (arrow, enlarged in *Figure 6h*) inside the receptacle. PB22226. (G) Side view of a flower bud (Flower 1 in *Figure 2b*) with longitudinal ribs (arrows) on the sepals (s) and petals (p). PB22223. (H) Side view of Flower 1 in *Figure 1d*, showing a receptacle (h), perianth (black arrows), and a dendroid style (white arrow). PB22224. (I) Side view of Flower 15 in *Figure 1e*, without sepals or petals. PB22222a. Bar = 1 mm. (J) Detailed view of the flower shown in *Figure 2e*, showing the arrangement of three petal bases (1-3) inside the sepals (s). These petals bases correspond to the three petals (1-3) in *Figure 2e*. PB22278.
DOI: https://doi.org/10.7554/eLife.38827.005

The following figure supplement is available for figure 2:

**Figure supplement 1.** Frequently observed palynomorphs associated with *Nanjinganthus*.
DOI: https://doi.org/10.7554/eLife.38827.006

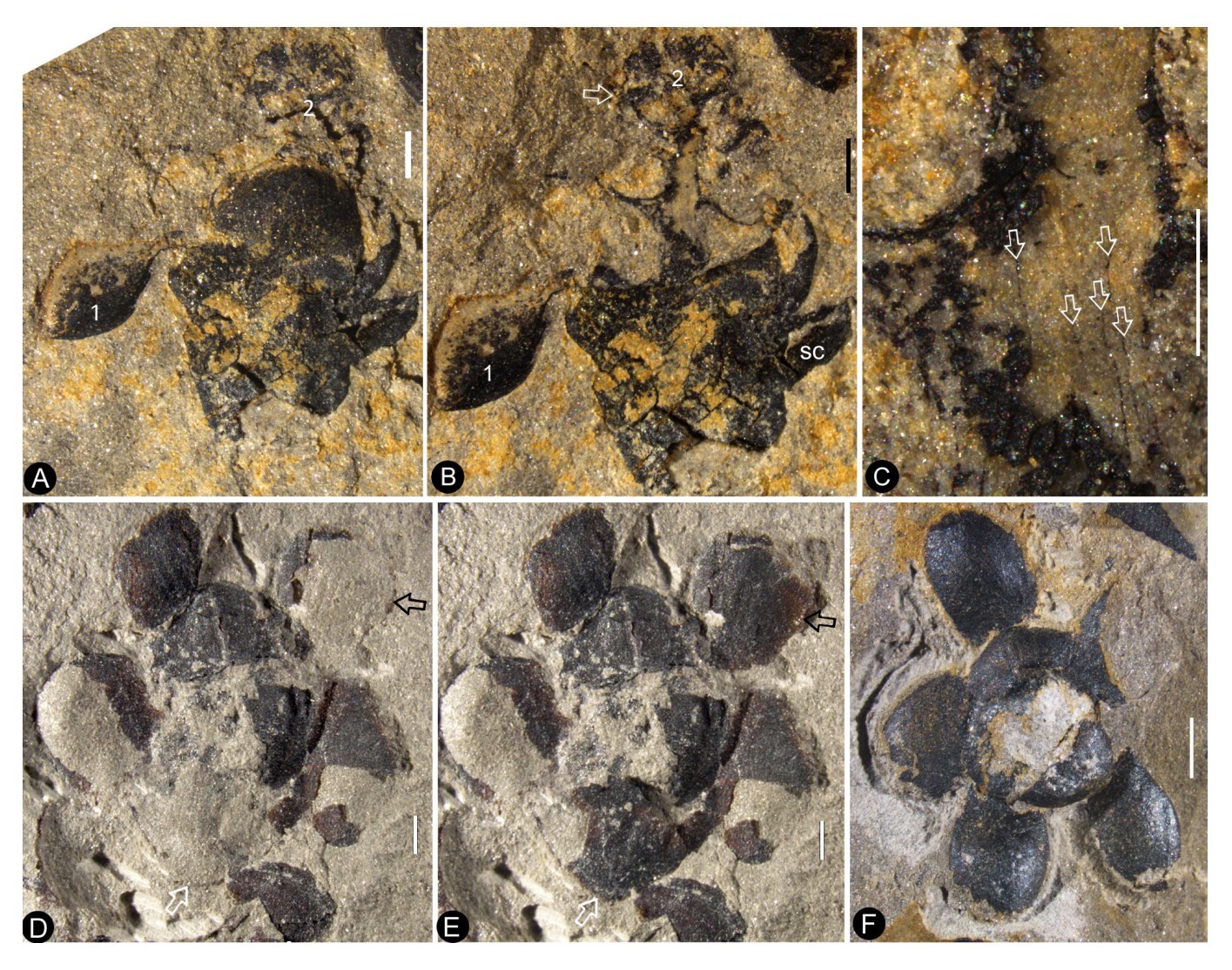

**Figure 3.** Individuals of *Nanjinganthus*. Bar = 1 mm except otherwise annotated. (A–C, PB22224) (A) Flower 2 in *Figure 1d* (before the dégagement), showing the petal (1) and style (2) still embedded in the sediments. (B The same flower as in *Figure 3a*, after dégagement, showing the exposed dendroid style (white arrow) and petal (1), and the scale (sc) on the side of receptacle. (C) Detailed view of the style shown in *Figure 3b* with faint striations (arrows). Bar = 0.5 mm. (D–E) Flower 2 in *Figure 2b* after and before the organic material of the sepals (white arrows) and petals (black arrows) were removed for cuticle analysis. PB22223. (F) Bottom view of a flower before processing. Internal details are shown in *Figure 2e,j*. PB22278.
DOI: https://doi.org/10.7554/eLife.38827.007

The following figure supplement is available for figure 3:

**Figure supplement 1.** Fossil plants associated with *Nanjinganthus*.
DOI: https://doi.org/10.7554/eLife.38827.008

branches appear oppositely arranged along the style (*Figures 2h* and *3b*) while the upper ones appear irregularly arranged (*Figures 2i*, *6a* and *7c–d*). There are longitudinal faint striations on the surface of the style (*Figures 3c* and *5j*). There are 1.6–3.6 mm long and 1.7–2.2 mm wide round-tri-angular scales on the sides of the ovary (*Figures 2g*, *3b*, *4a,g–h* and *5i*). Each ovary contains one to three seeds that are 0.65–3 mm x 0.5–1.7 mm, elongated or oval-shaped (*Figures 2f*, *5a,c* and *6d,f–l*), hanged on the inner wall of the ovary by a 0.08–0.27 mm wide funiculus (*Figures 5e* and *6d–e*). A micropyle-like depression 0.15 × 0.36 mm is seen on a seed (*Figure 5a,d*).

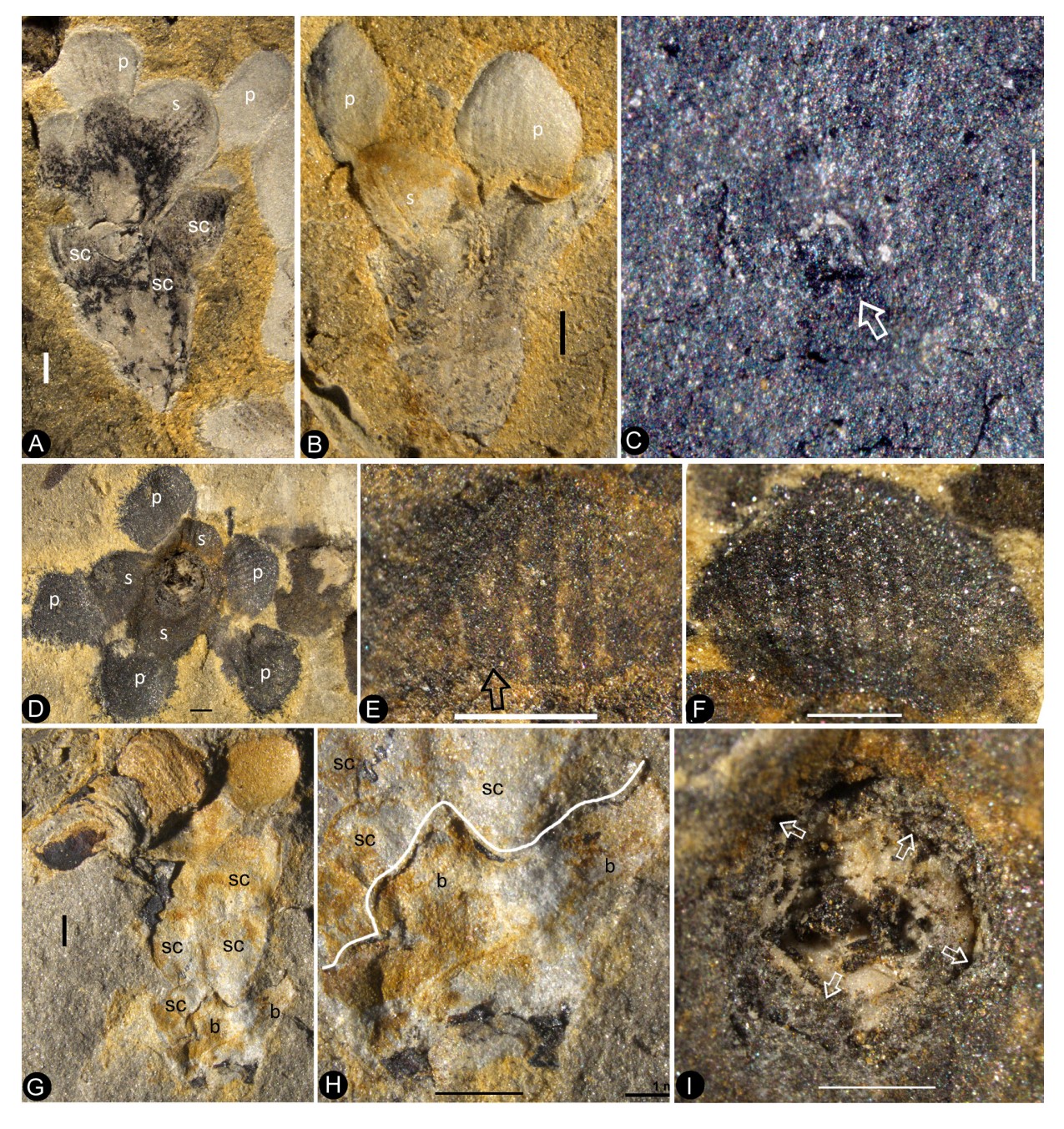

**Figure 4.** *Nanjinganthus* flowers preserved in various orientations and states. Bar = 1 mm except otherwise annotated. (**A**) An oblique longitudinally split flower (Flower 11 in *Figure 1e*) with scales (sc), sepals (s), and petals (p). PB22222a. (**B**) A longitudinally split flower (Flower 12 in *Figure 1e*) with sepals (s) and petals (p). PB22222a. (**C**) Integral surface of an ovarian roof with a scar (arrow) left by a broken off style, from the flower shown in *Figure 5h*. PB22279. Bar = 0.5 mm. (**D**) Bottom view of a flower (Flower 14 in *Figure 1e*) with three sepals (s) and five petals (p) visible. PB22222a. (**E**) One of the sepals in *Figure 4d*, showing longitudinal ribs forking (arrow). PB22222a. (**F**) One of the petals in *Figure 4d*, showing longitudinal ribs. PB22222a. (**G**) Side view of a flower, showing scales (sc) on the ovary side and connate bracts (b) at the bottom. PB22229. (**H**) Detailed view of the connate bracts (b) and scales (sc) in *Figure 4g*. Note the outline (white line) of the fused bracts. PB22229. (**I**) The locule surrounded by the ovary wall (arrows) of the flower shown in *Figure 4d*. PB22222a.

DOI: https://doi.org/10.7554/eLife.38827.009

The following figure supplement is available for figure 4:

**Figure supplement 1.** Fossil plants associated with *Nanjinganthus*.
DOI: https://doi.org/10.7554/eLife.38827.010

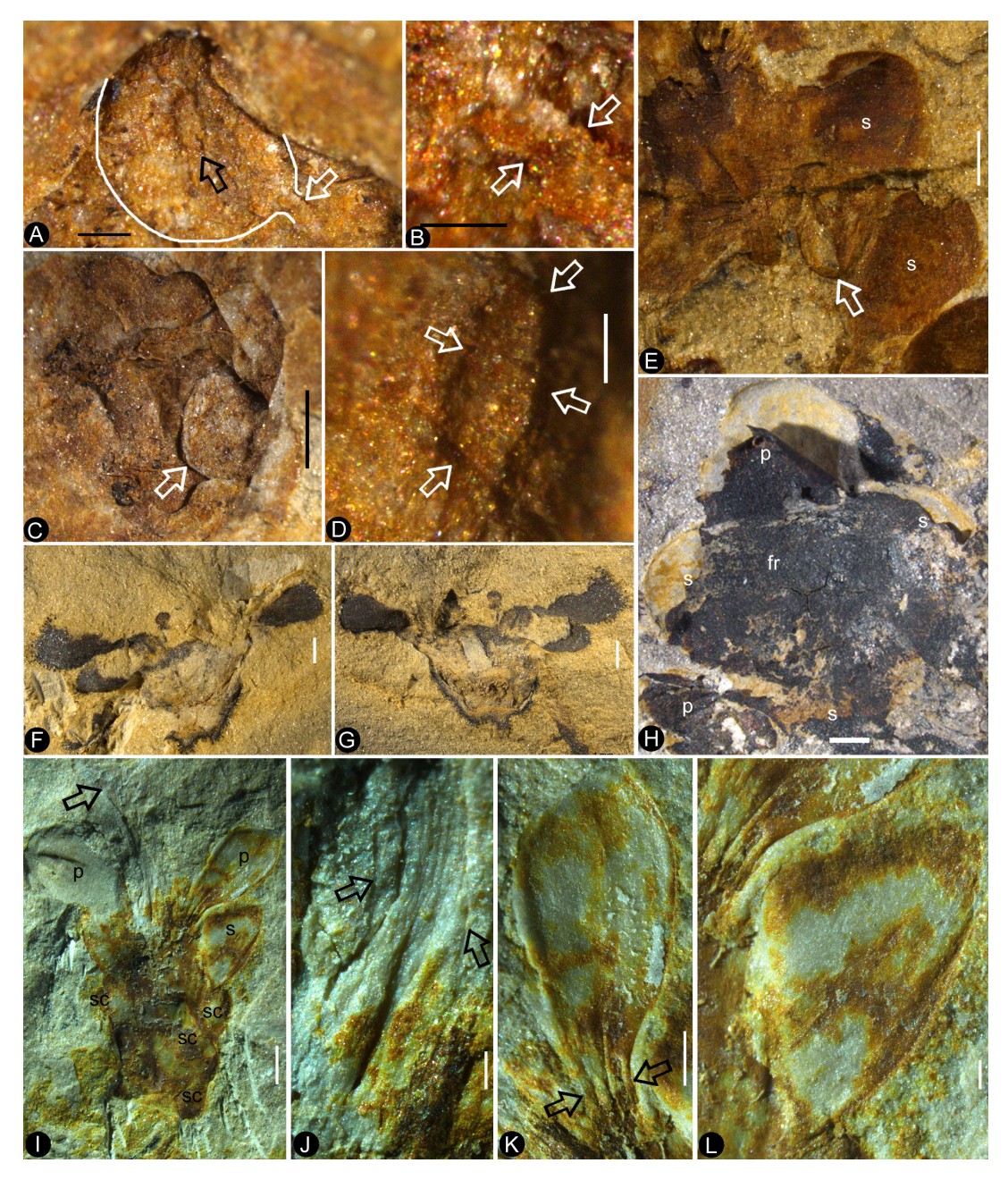

**Figure 5.** *In situ* seeds and flowers.  Bar = 1 mm except otherwise annotated. (**A**) A seed (outlined) inside the ovary of Flower 16 in **Figure 1e**. Note the oboval micropyle (black arrow) and funiculus (white arrow). PB22222a. Bar = 0.2 mm. (**B**) Detailed view of the funiculus (between the arrows) of the seed in **Figure 5a**. PB22222a. Bar = 0.1 mm. (**C**) A seed (detailed in **Figure 6i**) inside the ovary of Flower 7 in **Figure 1e**. PB22222a. (**D**) Detailed view of the oval micropyle (arrows) of the seed in **Figure 5a**. PB22222a. Bar = 0.1 mm. (**E**) A seed (arrow, detailed in **Figure 6d–e**) inside the receptacle in Flower two in **Figure 1b**. PB22226. (**F, G**) Two facing parts of the same flower (Flower 10 in **Figure 1e**). PB22222a. (**H**) Top view of a flower with organically-preserved sepals (s), petals (p) and integral ovarian roof (fr), which is detailed in **Figure 4c**. PB22279. (**I**) Side view of a longitudinally split flower with scales (sc) on ovary side, sepals (s), petals (p) and partially preserved style (arrow). PB22489. (**J**) Detailed view of basal portion of the style (between arrows) arrowed in **Figure 5i**, with faint longitudinal striations. PB22489. Bar = 0.2 mm. (**K**) Detailed view of the narrowing base (between arrows) of the right petal in **Figure 5i**. PB22489. Bar = 0.5 mm.( **L**) Detailed view of a sepal in **Figure 5i**. PB22489. Bar = 0.2 mm.

DOI: https://doi.org/10.7554/eLife.38827.011

The following figure supplement is available for figure 5:

**Figure supplement 1.** Flowers of a living angiosperm and its details.

DOI: https://doi.org/10.7554/eLife.38827.012

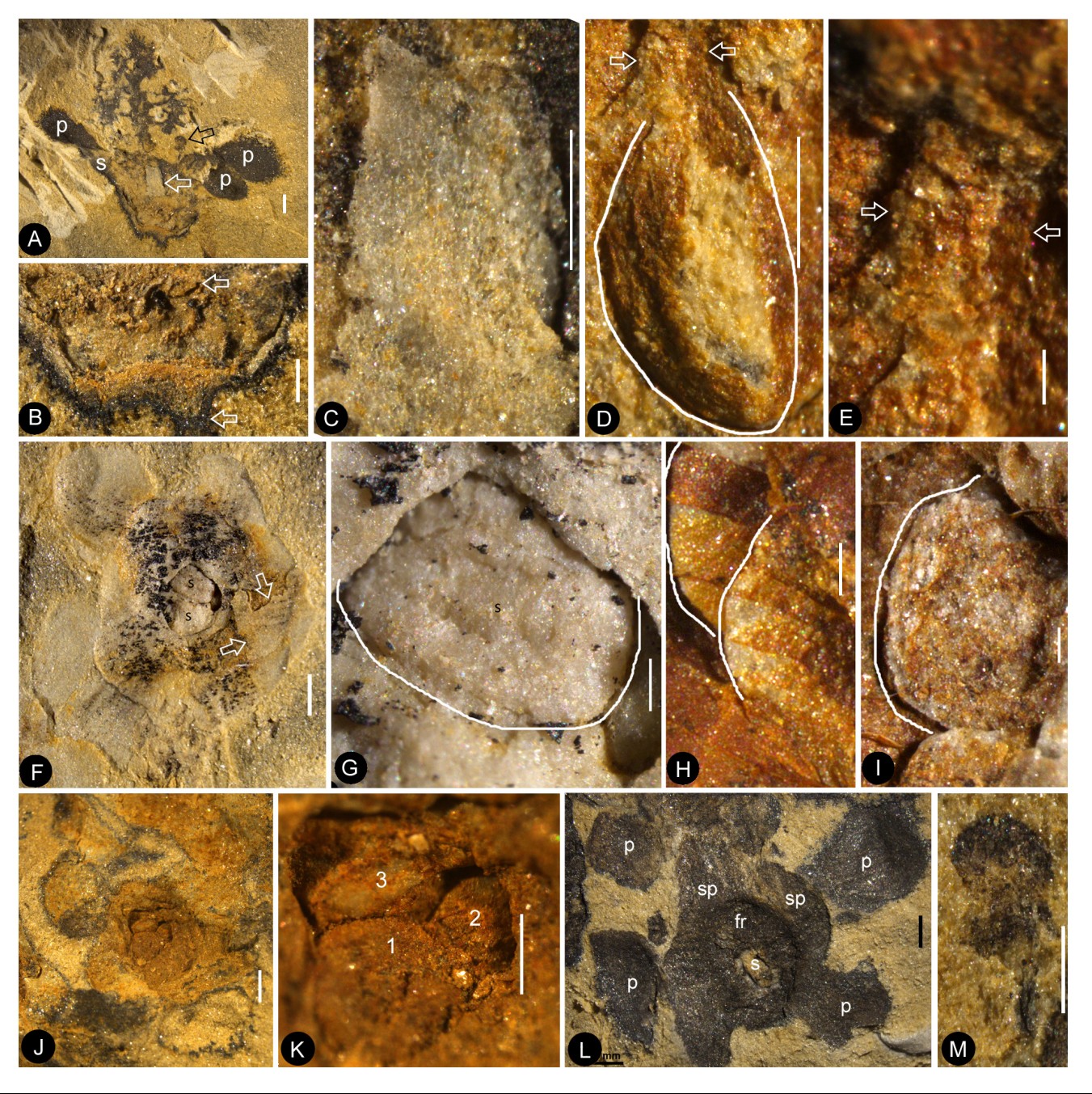

**Figure 6.** Dendroid style, *in situ* seeds, and details of flowers. PB22222a, Bar = 1 mm except otherwise annotated. (**A**) A longitudinally split flower (counterpart of Flower 10 in *Figure 1e*, the same as in *Figure 5f–g*) showing the sepal (s) and petals (p), style base (white arrow), and an unknown organ (black arrow). (**B**) Detailed view showing the pedicel (lower arrow) terminating at the bottom of the ovary in *Figure 6a*. Note the level of ovarian roof (upper arrow). Bar = 0.5 mm. (**C**) Detailed view of the basal portion of the style marked by white arrow in *Figure 6a*. Bar = 0.5 mm. (**D**) A seed (white line) hanging by its funiculus (between arrows) on the ovarian wall of the Flower 2 in *Figure 1b*. PB22226. Bar = 0.5 mm. (**E**) Detailed view of the funiculus (between arrows) of the seed in *Figure 6d*. PB22226. Bar = 0.1 mm.( **F**) Top view of Flower 8 in *Figure 1e* with sepals and petals surrounding the ovary containing two seeds (s). Note the residue (arrows) of the ovarian roof. (**G**) Detailed view of one of the oval seeds (s) inside the ovary in *Figure 6f*. Bar = 0.2 mm. (**H**) Two seeds (white line), one overlapping the other, inside the ovary shown in *Figure 2f*. PB22226. Bar = 0.2 mm. (**I**) An oval seed (white line) inside the ovary of Flower 7 in *Figure 1e*. Bar = 0.2 mm. (**J**) Detailed view of Flower 1 in *Figure 1g*, showing seeds within ovary. PB22228. (**K**) Detailed view of three seeds (1-3) inside the ovary of the flower shown in *Figure 6j*. PB22228. Bar = 0.5 mm. (**L**) Top view of a flower showing petals (p), sepal (sp), seed (s) visible under the ovarian roof (fr). PB22222d. (**M**) Detailed view of the unknown organ (staminode?) marked by the black arrow in *Figure 6a*. Bar = 0.5 mm.

DOI: https://doi.org/10.7554/eLife.38827.013

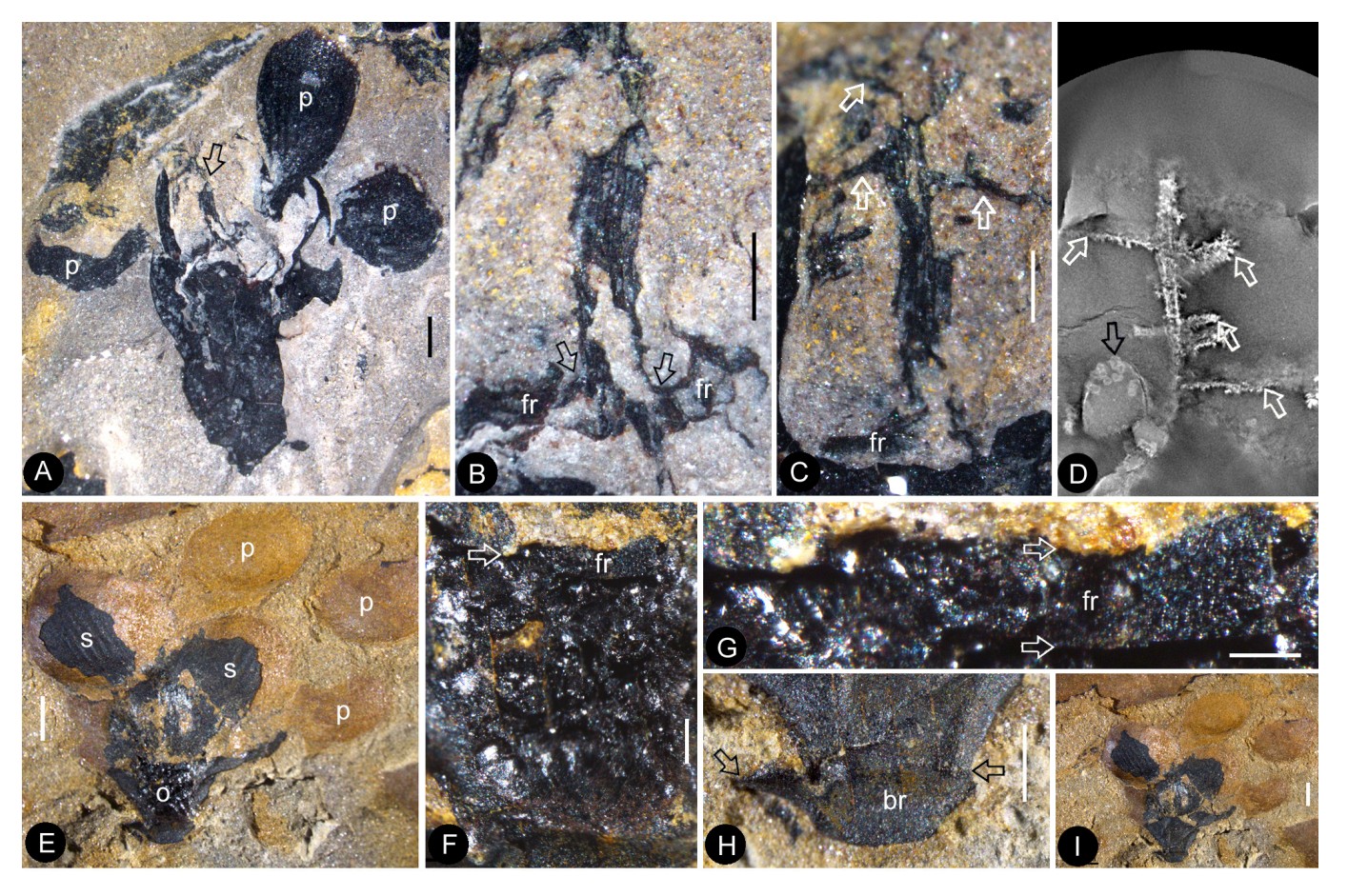

**Figure 7.** The flowers and their internal details. (A–C) (E-I) stereomicroscopy; (D), micro-CL. Bar = 1 mm except otherwise annotated. (A) A flower carefully dégaged to expose the details of the gynoecium. Note the petals (p) and a style (arrow) in the center. PB22282. (B) Detailed view of the style in *Figure 7a*, showing its connection (arrows) to the ovarian roof (fr). PB22282. Bar = 0.5 mm. (C) Distal portion of the same style as in *Figure 7b*, showing its connection with the ovarian roof (fr) and dendroid form with lateral branches (arrows). PB22282. Bar = 0.5 mm. (D) Micro-CL slice 1169 showing a perianth element (black arrow) and branches (white arrows) of the style, embedded in sediments and thus invisible to naked eyes, of Flower 4 in *Figure 1e*. PB22222a. (E–I) PB22281. (E) Side view of an organically-preserved flower with sepals (s) and petals (p). Note the dark organic material in the ovary (o) and some sepals. The foreground portion of the receptacle has been removed (compare with *Figure 7i*), to show the details in *Figure 7f–h*. (F) Detailed view of the receptacle/ovary in *Figure 7e*. Note the ovarian roof (fr) preventing the outside (above) sediment (yellow color) from entering the ovarian locule. Bar = 0.2 mm. (G) Detailed view of the solid organically-preserved ovarian roof (fr) with integral outer (upper arrow) and inner (lower arrow) surfaces. Bar = 0.1 mm. (H) Bottom portion of the flower in *Figure 7i*, showing subtending bracts (br, arrows). Bar = 0.5 mm. (I) The flower in *Figure 7e*, before removing the foreground portion of the ovary.

DOI: https://doi.org/10.7554/eLife.38827.014

## Holotype
*Figure 2d* (PB22222B).

## Isotypes
*Figure 6a,f* (PB22222a), *Figure 7e–i* (PB22281), *Figure 5h* (PB22279).

## Specimens
PB22222-PB22229, PB22236, PB22238, PB22241-PB22243, PB22245-PB22247, PB22256-PB22260, PB22278-PB22282, PB22489.

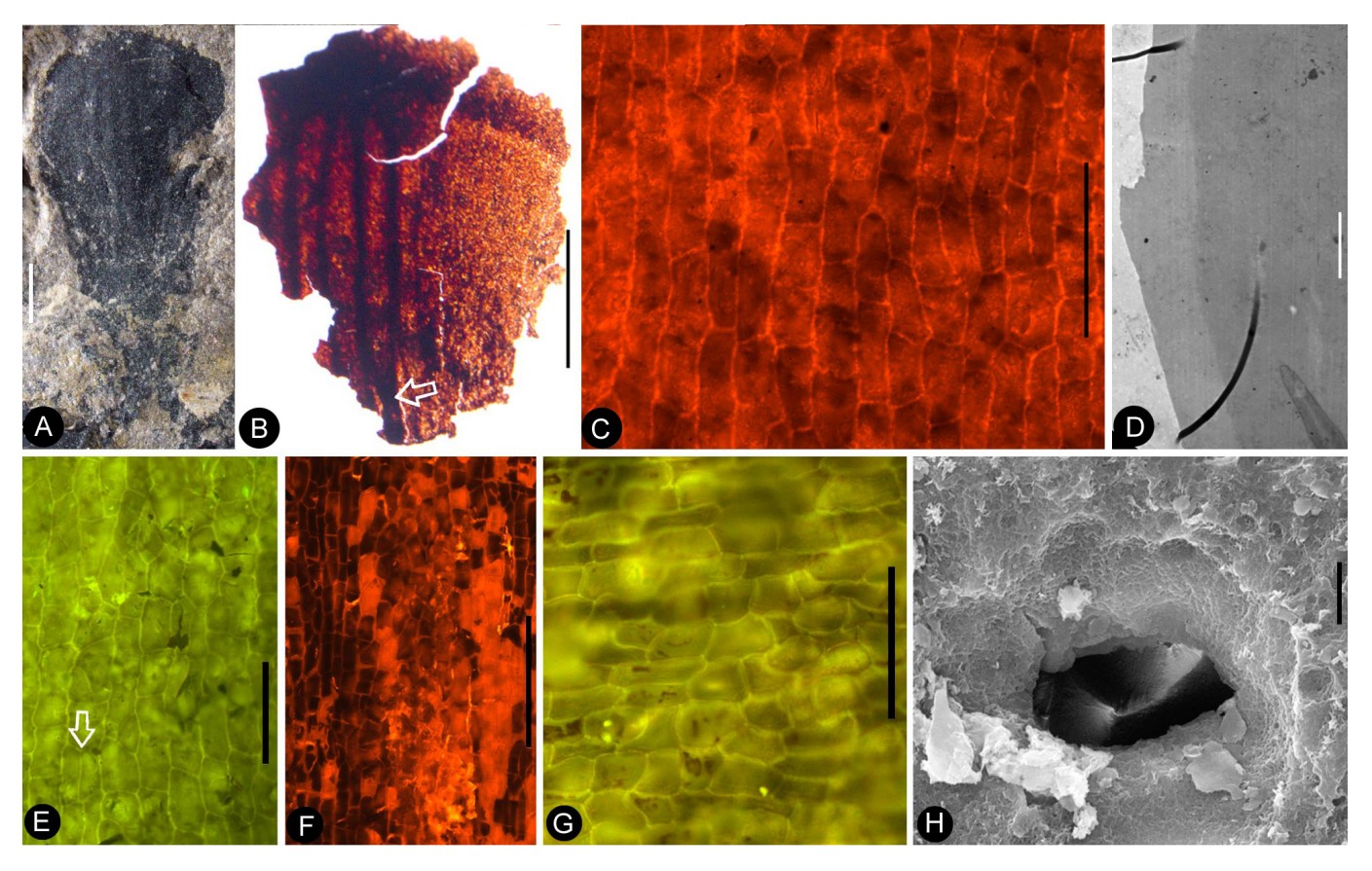

**Figure 8.** Details of the sepal and petal. (A-B) stereomicroscopy; (C) (E-G) fluorescence light microscopy; (D) TEM; (H) SEM. Bar = 1 mm except otherwise annotated. (**A**) A petal with a narrowing base. PB22280. (**B**) A partial petal from the Flower in *Figure 3d–e*, with the longitudinal rib (to the left) forking at the base (arrow) and the rib-free laminar area to the right. PB22223. (**C**) Elongated epidermal cells of the petal in Figure 8b . PB22223. Bar = 0.1 mm. (**D**) Transmission electron microscope view showing the cuticle (left, light color) of a petal. PB22223. Bar = 2 μm. (**E**) Elongated epidermal cells not in strict longitudinal files in the laminar portion of the petal in Figure 8b . Note the two newly formed epidermal cells (arrow). PB22223. Bar = 0.1 mm. (**F**) Ribs with elongated epidermal cells (left and right) alternating the between region with less elongated cells (middle) of the petal in Figure 8b . PB22223. Bar = 0.2 mm. (**G** Elongated (above) and isodiametric (below) epidermal cells on the sepal of Flower in *Figure 3d–e*. PB22223. Bar = 0.1 mm. (**H**) A stoma on the bract of the flower (marked by white arrow in *Figure 2h*). PB22224. Bar = 5 μm.

DOI: https://doi.org/10.7554/eLife.38827.015

## Etymology

*dendrostyla,* for 'tree-like' (*dendri-*) and 'style' (*-stylus*) in Latin.

## Remarks

The receptacle is 'the axis of a flower on which the perianth, androecium and gynoecium are borne' (*Stevens, 2018*). This is the definition followed here. The important characteristic of the receptacle in *Nanjinganthus* is its cup form, a form frequently seen in more derived angiosperms according to the APG system.

A dendroid style is seen in ten flowers (four in PB22224, *Figures 2h* and *3a–b*; four in PB2222a, *Figures 2i*, *5f–g*, *6a* and *7d*; one in PB22282, *Figure 7a–c*; one in PB22489, *Figure 5i–j*). The repeated occurrences of such an unexpected feature in the specimens of *Nanjinganthus* underscore its truthful existence. The dendroid-form distal portion of the gynoecium may be branched stigmas in *Nanjinganthus*. But it is possible that these lateral appendages on the style are actually pollen sac complexes, as are similarly attached on the style in extant Malvaceae (*Judd et al., 1999*). We have performed a meticulous fluorescence microscopic examination of this structure and found no trace of pollen grains, reducing the possibility that these lateral branches are clusters of pollen sacs, which

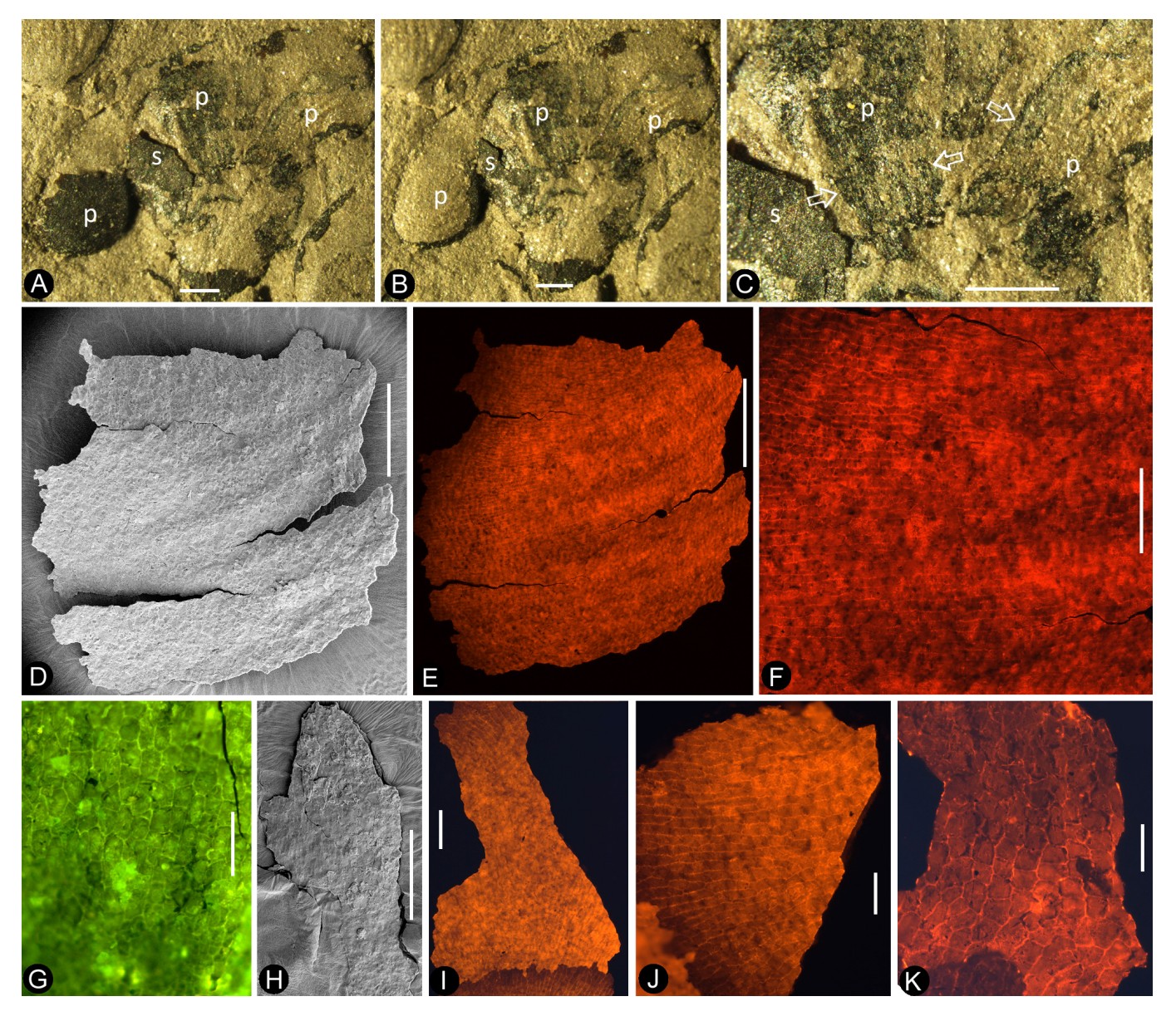

**Figure 9.** Petal and details of *Nanjinganthus*. (A-C) light stereomicroscopy; (D) (H) SEM; (E-G) (I-K) fluorescence light microscopy. PB22223. Bar = 1 mm except otherwise annotated. (A) Side view of Flower 3 in *Figure 2b*, showing the arrangement of the petals (p) and sepal (s). (B) The same flower as in *Figure 9a*. Note that some organic material of the petal has been removed for detailed observation. (C). Margins (arrows) of the petal (p) with cuneate base and their relationship to the sepal (s). (D) The petal removed from *Figure 9a*. SEM. Bar = 0.5 mm. (E) Cellular details of the petal in *Figure 9d* Bar = 0.5 mm. (F) Elongated epidermal cells arranged in files, enlarged from *Figure 9e*. Bar = 0.2 mm. (G) Isodiametric epidermal cells in the laminar area portion of the petal in *Figure 9e*. Bar = 0.1 mm. (H) A fragment of the sepal seen in *Figure 9a*. Bar = 0.5 mm. (I) Cellular details of the sepal in *Figure 9h*. Bar = 0.2 mm. (J) Elongated epidermal cells arranged in files on the sepal in *Figure 9i*. Bar = 0.1 mm. (K) Isodiametric epidermal cells on the laminar area of the sepal in *Figure 9a*. Bar = 0.1 mm.

DOI: https://doi.org/10.7554/eLife.38827.016

is the case seen in some angiosperms (Malvaceae). A branched distal projection is apparently lacking in all known gymnosperms, but it has been seen some derived angiosperms, such as Passifloraceae, Poaceae and Euphorbiaceae (*Heywood, 1978*). One of the advantages of a branched style is the increased receptive area, which is conducive to anemophilous pollination. The occurrence of such feature in *Nanjinganthus* might suggest that *Nanjinganthus* had yet not established a close

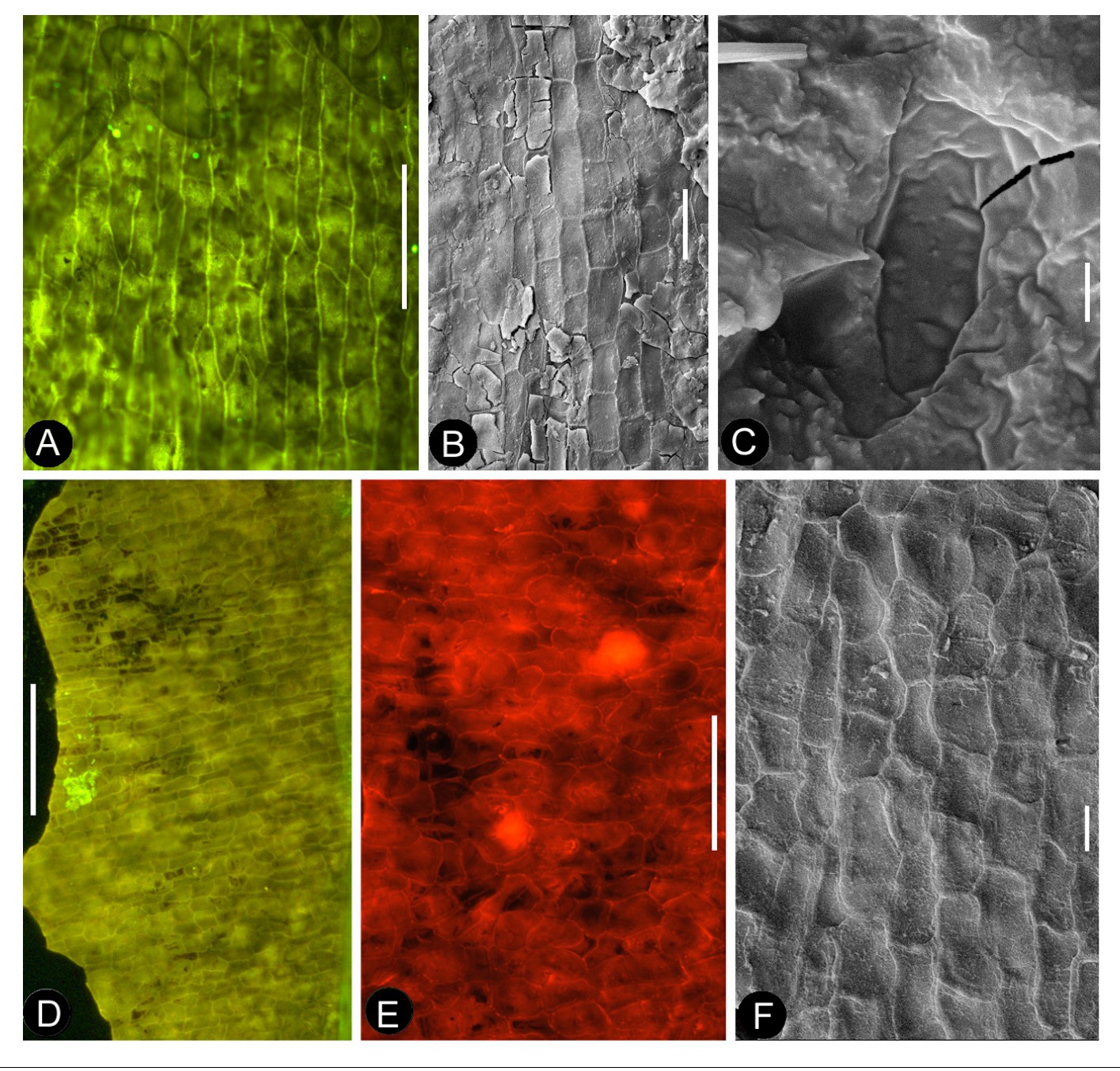

**Figure 10.** Cuticular details of *Nanjinganthus*. A, D-E, Fluorescence light microscopy; B-C, F, SEM. PB22223. (**A**) Elongated epidermal cells in longitudinal files in the middle portion of the petal in Figure 8b. Bar = 0.1 mm. (**B**) Elongated epidermal cells on the rib of the petal in Figure 8b. Bar = 50 μm. (**C**) A possible stoma on the petal shown in Figure 8b. Bar = 2 μm. (**D**) Elongated epidermal cells in files on the sepal of flower in *Figure 3d–e*. Bar = 0.2 mm. (**E**) Isodiametric epidermal cells on the sepal of flower in *Figure 3d–e*. Bar = 0.1 mm. (**F**) Isodiametric epidermal cells on the sepal of flower in *Figure 3d–e*. Bar = 20 μm.

DOI: https://doi.org/10.7554/eLife.38827.017

cooperation with animals (insects). However, it is noteworthy that this feature is not seen among extant basal angiosperms *sensu* APG (*Chase et al., 2016*). Considering the extremely early age of *Nanjinganthus*, we refrain from correlating *Nanjinganthus* with assumed derived taxa

(Malvaceae and Rosaceae). We hope the future research may shed more light the nature of this part of *Nanjinganthus*.

We have not seen any trace of the carpels typical of Magnoliales, which were previously believed by some to represent ancestral angiosperms. The seeds are physically enclosed by the cup-form receptacle and ovarian roof in *Nanjinganthus*. This constitutes the foundation based on which we justify our interpretation of *Nanjinganthus* as an angiosperm. The lack of carpel typical of Magnoliales cannot prevent *Nanjinganthus* from being an angiosperm as many angiosperms are actually 'acarpellate' (*Heads, 1984*; *Sattler and Lacroix, 1988*). It is noteworthy that, at least in some of basal angiosperms such as *Nymphaea* (Nymphaeales) (*Taylor, 1991*; *Taylor, 1996*) and derived angiosperms such as Cactaceae (*Boke, 1964*), the ovary is inferior and the seeds are attached to the ovarian walls. Whether the ovaries in these taxa share similar derivation pathway is a question worthy of further investigation.

Four terms are used to describe the foliar parts in *Nanjinganthus*, namely, bract, scale, sepal, and petal. These terms are used according to the following demarcations and definitions. Bracts designate the foliar parts subtending the ovary. The scales are the foliar parts attached to the sides of the ovary. The sepals are those foliar parts attached to the rim of the receptacle with their whole bases. And the petals are foliar parts with narrowing bases attached to the receptacle rim and inside the sepals. Similar occurrence of bracts, sepals and petals is seen in some extant flowers (*Figure 5—figure supplement 1*).

The enclosure of the seeds is fulfilled by the cup-form receptacle from the bottom and the structure here-called 'ovarian roof' (preserved complete in *Figures 4c*, *5h* and *7e–g*, but partially preserved in *Figures 2f*, *5c* and *6f,j,l*) from the above. The intact ovarian roof is clearly seen in the side view (*Figure 7f–g*) and in surface view (*Figures 4c* and *5h*), in the latter case the seeds inside ovary are fully eclipsed by the ovarian roof. The ovarian roof is partially lost in *Figure 6l*, in which a central portion of the ovarian roof broke off revealing one of the seeds inside the ovary. The ovarian roof is almost completely lost (but still with some of its residue) in *Figure 6f*, and finally fully lost in *Figures 2f* and *6j–k*, in which the seeds are plainly visible. This series of varying preservation status of ovarian roof suggests that the ovarian roof has fully enclosed the ovules in its original status, and the loss of ovarian roof and exposure of seeds are artifacts due to preservation.

We cannot recognize the maturity of the ovules/seeds in *Nanjinganthus*, the length about 1 mm suggests that they are most likely to be seeds rather than ovules, therefore we prefer to use the term 'seed' rather 'ovule' throughout this paper. The number of seeds in *Nanjinganthus* is variable. According to our observation, it may be one (not shown), two (*Figure 6f–i*), or even three (*Figure 6j–k*).

## Discussion

### Alternative interpretations

The Mesozoic was an age of gymnosperms, so the Jurassic age of *Nanjinganthus* makes it necessary to compare *Nanjinganthus* with common fossil gymnosperms frequently seen in the Mesozoic first. The potential candidates for *Nanjinganthus* include Caytoniales, Corystospermales, Ginkgoales, Czekanowskiales, Coniferales, Iraniales, Pentoxyales, Bennettitales, and Gnetales.

*Caytonia* is a very intriguing fossil plant that has been frequently compared with angiosperms (*Thomas, 1925*; *Doyle, 2006*). Regardless of its ultimate gymnospermous affinity (*Thomas, 1925*; *Harris, 1933*; *Harris, 1940*; *Nixon et al., 1994*), *Caytonia* can be easily distinguished from *Nanjinganthus* by its cupule with an adaxial basal opening, bilateral reproductive organs, and lack of both a dendroid style and foliar appendages in its reproductive organs.

Corystospermales is usually considered as a Mesozoic seed fern group, unlike *Caytonia*, the cupules in most Corystospermales open on the abaxial base and are rarely compared with angiosperms (*Taylor et al., 2009*). Corystospermales can be easily distinguished from *Nanjinganthus* by their cupule which has an abaxial basal opening, bilateral reproductive organs, and the lack of both a dendroid style and foliar appendages in the reproductive organs.

Ginkgoales diversified greatly during the Mesozoic, and unlike extant *Ginkgo*, the Mesozoic relatives of *Ginkgo* are well represented by their reproductive organs, which are composed of seeds in

clusters (*Zhou and Zheng, 2003*). Ginkgoales can be easily distinguished from *Nanjinganthus* by their clustered naked seeds, and lack of a dendroid style in the reproductive organs.

Czekanowskiales are a unique group of fossil plants restricted to the Mesozoic. Their reproductive organs are bivalvate cupules containing two rows of seeds. Czekanowskiales can be easily distinguished from *Nanjinganthus* by their bivalvate cupules, bilateral reproductive organs, and lack of both a dendroid style and foliar appendages in the reproductive organs.

*Irania* is the single genus of the Iraniales, which is assumed to have borne clusters of pollen sacs and fruits, from the Triassic-Jurassic (*Schweitzer, 1977*). Although no seeds have been observed in *Irania*, it is suspected to be an angiosperm. Its female and male parts are not concentrated on the same axis, and do not constitute a flower-like structure, and it is unknown whether the seeds are enclosed. These features distinguish *Irania* from *Nanjinganthus*.

Pentoxylales are Mesozoic woody fossil plants characterized by a stem with five steles (*Taylor et al., 2009*). Their reproductive organs are cones with numerous naked orthotropous seeds helically arranged around the axes of their cones. Pentoxylales can be easily distinguished from *Nanjinganthus* by their cones which are devoid of any foliar appendages and the lack of a dendroid style.

Bennettitales are important Mesozoic gymnosperms that are frequently related to angiosperms (*Crane, 1985*; *Rothwell et al., 2009*). Their reproductive organs are characterized by orthotropous ovules with micropylar tubes dispersed among interseminal scales, and these parts are helically arranged along the cone axis. Bennettitales can be easily distinguished from *Nanjinganthus* by their cones with ovules bearing micropylar tubes, and lack of a dendroid style (*Taylor et al., 2009*).

Gnetales are important gymnosperms that diversified once in the Mesozoic, among which *Gnetum* has leaves that are difficult to distinguish from eudicots (*Martens, 1971*; *Biswas and Johri, 1997*). A characteristic feature of Gnetales is their decussate arrangement of leaves and cone parts. Like in Bennettitales, the reproductive organs of Gnetales are characterized by orthotropous ovules with micropylar tubes surrounded by scales. Like Bennettitales, Gnetales can be easily distinguished from *Nanjinganthus* by their cones with ovules with micropylar tubes and lack of a dendroid style.

Besides the above comparison among female organs of seed plants, it is noteworthy that male cones in some conifers demonstrate a certain resemblance to *Nanjinganthus*, although such a comparison appears spurious when the presence of seeds in *Nanjinganthus* is taken into consideration. The bud-like *Nanjinganthus* (*Figures 2g*, *4a–b,g* and *5i*) appears similar to male cones of *Microbiota decussata* (a3 in Figure 2 of *Schulz et al., 2014*) and *Thuja plicata* (f2 in Figure 2 of *Schulz et al., 2014*). Mature *Nanjinganthus* (*Figures 2h–i*, *3a–b*, *5f–g*, *6a* and *7a,e,i*) appear like the male cone of *Sequoia sempervirens, Taxus floridana,* and *Tsuga canadensis* (e2, e5, f4, respectively, in Figure 2 of *Schulz et al., 2014*). The sporangiophores in these taxa are arranged around the cone axis and appear dendrioid, and the scales at the base appear like the sepals/petals in *Nanjinganthus*. But the cup-form receptacle with seeds inside plus the lack of pollen grains in the distal dendroid part distinguish *Nanjinganthus* from all these male cones. As in female cones, these male cones also have cone axes penetrating the cones from the bottom to the tip and thus are different from *Nanjinganthus* in which the pedicel stops at the bottom of the organ (ovary) (*Figure 6b*) and the style starts above the ovarian roof (*Figure 4c*, *5h*, *6a-c*, *7a-c*). In addition, the spatial distribution and furcation of the vascular bundles in the sepals and petals of *Nanjinganthus* (*Figures 2c–d,4e–f,9a–b*) are distinct from those seen in bracts and scales in coniferous cones.

From the above comparison, we infer that none of the known gymnosperms, fossil or extant, are comparable to *Nanjinganthus*. The enclosed seeds distinguish *Nanjinganthus* from gymnosperms, which are not supposed to enclose their ovules in such a way (*Table 1*).

There are several reports of Jurassic angiosperms, including *Schmeissneria* (*Wang et al., 2007*), *Xingxueanthus* (*Wang and Wang, 2010*), *Juraherba* (*Han et al., 2016*), and *Euanthus* (*Liu and Wang, 2016*). These genera are from the Middle-Late Jurassic of northeastern China. The cup-form receptacle, inferior ovary, and dendroid style distinguish *Nanjinganthus* from all these Jurassic peers, and justify *Nanjinganthus* as a new genus of angiosperm.

Although multiple characters have been suggested to identify fossil angiosperms (*Herendeen et al., 2017*), angio-ovuly before pollination is the key character that distinguishes an angiosperm from other seed plants (*Tomlinson and Takaso, 2002*; *Wang, 2010a*; *Wang, 2018*). This criterion has been repeatedly applied to identify fossil angiosperms (i.e. *Archaefructus* (*Sun et al., 1998*), which initially had no other features (stamen, venation, pollen grains) but enclosed

**Table 1.** Comparison between *Nanjinganthus* and Mesozoic gymnosperms.

| | *Nanjinganthus* | *Caytonia* | Bennettitales | Corystospermales | Ginkgoales | Coniferales | Iraniales | Czekanowskiales | Pentoxylales |
|---|---|---|---|---|---|---|---|---|---|
| Symmetry | Radial | Bilateral | Radial | Bilateral | Radial | Radial | Radial? | Bilateral | Radial? |
| With foliar appendages | Yes | No | No | No | No | ? | No | No | No |
| Enclosed seed | Yes | No | No | No | No | No? | ? | No | No |
| Opening in female part | No | Adaxial basal | Terminal? | Adaxial basal | N/A | N/A | ? | Distal slit | ? |
| Penetrating cone axis | No | Yes | Yes | Yes | ? | Yes | ? | Yes | Yes |

DOI: https://doi.org/10.7554/eLife.38827.018

seeds to support their angiospermous affinity). The integral ovarian roof of *Nanjinganthus* has no opening (*Figures 4c* and *5h*). After burial, this ovarian roof can block the sediment from entering the ovarian locule (*Figure 7e–g*). That this space remained free of sediment suggests a full enclosure of the ovules/seeds, securing an angiospermous affinity for *Nanjinganthus*.

The radial arrangement of two whorls of foliar parts (sepals and petals) in *Nanjinganthus* is very similar to those of flowers in extant angiosperms (*Figures 1a–b* and *2a–f*), while the above comparison with known gymnosperms emphasizes that *Nanjinganthus* cannot be interpreted as a gymnosperm. Furthermore, *Nanjinganthus* satisfies all thirteen definitions of flowers advanced by various authors (*Bateman et al., 2006*). These features again are consistent with the angiospermous affinity of *Nanjinganthus* (*Figure 11*).

There have been several suggested models of ancestral angiosperms (*Arber and Parkin, 1907*; *Cronquist, 1988*; *Endress and Doyle, 2015*; *Sauquet et al., 2017*). These models were drawn more or less after the assumed basalmost living angiosperms, either *Magnolia* or *Amborella*. The common features of these model plants include apocarpy, superior ovary, lack of obvious style, etc. However, these features are rarely seen in *Nanjinganthus* or other early angiosperms (*Wang et al., 2007*; *Wang and Zheng, 2009*; *Wang, 2010b*; *Wang, 2018*; *Han et al., 2013*; *Han et al., 2016*; *Han et al., 2017*; *Liu and Wang, 2016*; *Liu and Wang, 2017*; *Liu et al., 2018*; *Liu and Wang, 2018*). Instead, an inferior ovary, a feature unexpected by, at least, most theories of angiosperm evolution, is clearly seen in *Nanjinganthus* and quite many Early Cretaceous flowers (*Friis, 1984*; *Friis, 1990*; *Friis et al., 2011*). This discrepancy between fossil observation and theories suggests EITHER that inferences based on living plants have limited capability of 'predicting' past history, OR that angiosperms originated polyphyletically, each lineage has followed a different evolution route, and *Nanjinganthus* represents one of the many, OR that angiosperms have a history that dates back to a time much earlier than the Cretaceous, and *Nanjinganthus* is one of the many derived from such assumed ancestor, OR a combination of these. Whatever the implications are, the currently dominant theories of angiosperm evolution apparently need to be reassessed.

Most *Nanjinganthus* specimens are concentrated on certain bedding surfaces, and over 50 individual flowers are preserved on a single slab (*Figures 1a–g* and *2a–b*), suggesting that *Nanjinganthus* may have flourished and dominated a particular niche, although *Nanjinganthus* played only an inferior role in the broader Jurassic ecosystem. The concentrated preservation of delicate flowers is more likely a result of autochthonous preservation, suggesting a habitat very close to water for *Nanjinganthus*.

Various studies (including palaeobotany) on the South Xiangshan Formation in the last century (*Hsieh, 1928*; *Li et al., 1935*; *Sze and Chow, 1962*; *Zhou and Li, 1980*; *Cao, 1982*; *Cao, 1998*; *Cao, 2000*; *Wang et al., 1982*; *Huang, 1983*; *Huang, 1988*; *Ju, 1987*) and our palynological analysis as well as U/Pb dating (*Figure 1—figure supplement 1*; *Supplementary file 2*; *Santos et al., 2018*) suggest a latest Early Jurassic age for *Nanjinganthus*. Together with the 'unexpectedly' great diversity of angiosperms in the Early Cretaceous (*Sun et al., 1998*; *Sun et al., 2001*; *Sun et al., 2002*; *Leng and Friis, 2003*; *Leng and Friis, 2006*; *Ji et al., 2004*; *Wang and Zheng, 2009*; *Wang and Zheng, 2012*; *Wang and Han, 2011*; *Han et al., 2013*; *Han et al., 2017*; *Wang, 2015*; *Liu et al., 2010a*), pollen grains indistinguishable from angiosperms in the Triassic (*Hochuli and Feist-Burkhardt, 2004*; *Hochuli and Feist-Burkhardt, 2013*), a bisexual flower from the Jurassic

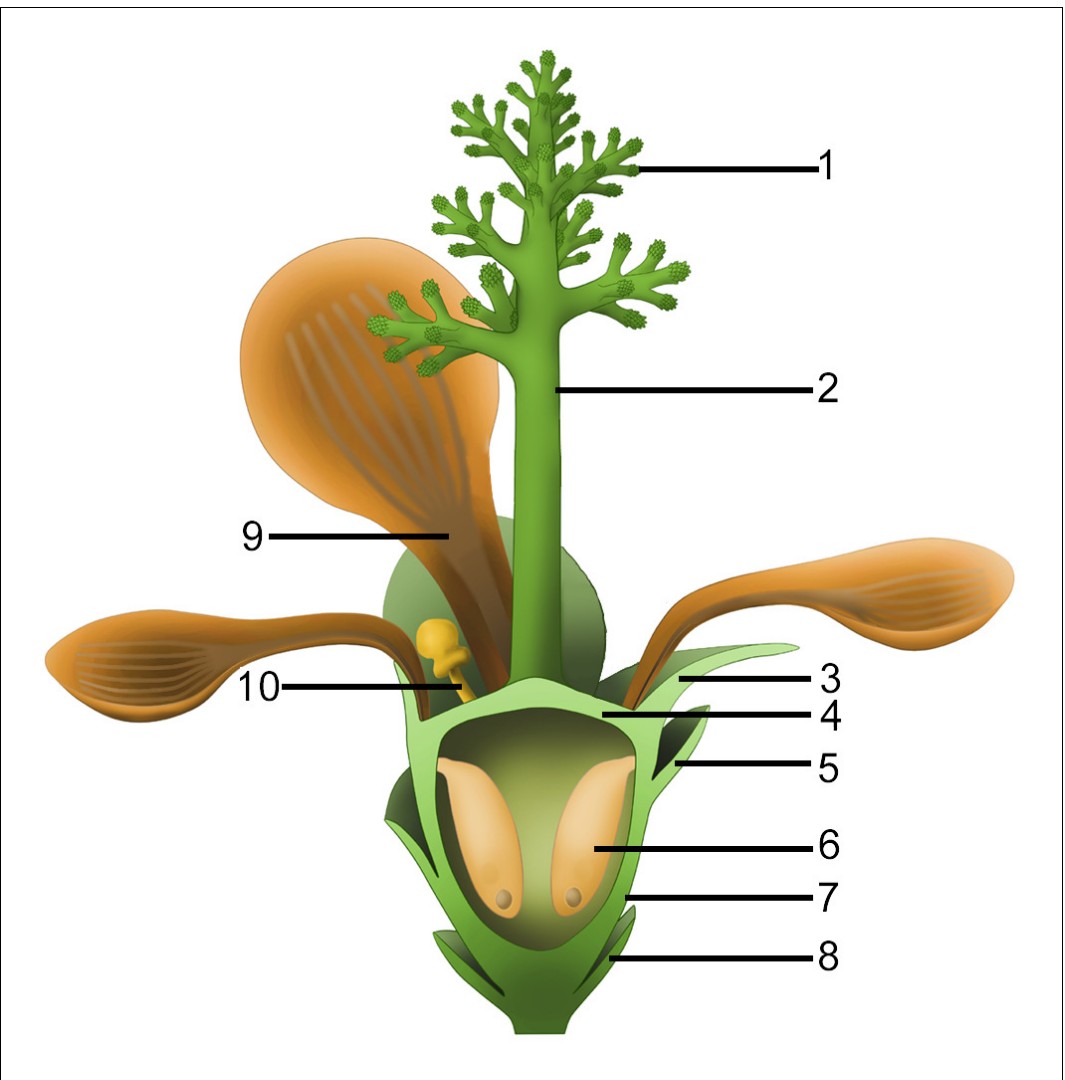

**Figure 11.** Idealized reconstruction of *Nanjinganthus*. 1, branches of dendroid style; 2, dendroid style; 3, sepal; 4, ovarian roof; 5, scale; 6, seed; 7, cup-form receptacle/ovary; 8, bract; 9, petal; 10, unknown organ (staminode?).
DOI: https://doi.org/10.7554/eLife.38827.019

(*Liu and Wang, 2016*), and an herbaceous angiosperm from the Middle Jurassic (*Han et al., 2016*), the unexpectedly early age of Poaceae (*Prasad et al., 2011*; *Wu et al., 2018*) and Solanaceae (*Wilf et al., 2017*), *Nanjinganthus* with over 200 specimens is consistent with a pre-Cretaceous origin of angiosperms.

The systematic position of *Nanjinganthus* is now apparently open to further investigation, although it demonstrates a certain resemblance to Pentapetalae *sensu* (*Judd et al., 2016*). We cannot determine whether *Nanjinganthus* stands for a Jurassic stem group of angiosperms that started their radiation later in the Cretaceous or a lateral branch leading to an evolutionary dead end. It is premature to determine its phylogenetic position before more information of contemporaneous peers is available, although we welcome all phylogeneticists to evaluate *Nanjinganthus* in their own ways and perspectives.

## Conclusion

*Nanjinganthus* is recognized based on at least 198 individual flower fossils from the Early Jurassic that are preserved in various states and orientations. We infer that the seeds were enclosed by a cup-form receptacle and an ovarian roof, traits which suggest an angiospermous affinity for

*Nanjinganthus*. It would be intriguing to figure out in future research whether *Nanjinganthus* represents a stem group, a group derived from more ancient ancestors, or an evolutionary dead end of polyphyletic angiosperms. We hope that the discovery of *Nanjinganthus* will re-invigorate research on the origin and early history of angiosperms.

## Materials and methods

### Geological background

Initially, what is now known as the Xiangshan Group was called the 'Nanking Sandstein' that was put in the Upper Carboniferous by *Richthofen (1912)*. *Liu and Chao (1924)* thought that the Nanking Sandstein belonged to the Jurassic and renamed it the 'Tsung Shan Formation'. *Hsieh (1928)* subdivided the Tsungshan Formation into six units, namely, in ascending order, Huang Ma Ching Shale, Quartzitic Conglomerate, Tzu Hsia Tung Series, Lingkusssu Shale, Light Yellow Sandstone, Variegated Sandstone and Shale. *Li et al. (1935)* found fossil plants including *Equisetites*, *Neocalamites*, *Cladophlebis*, *Otozamites*, *Pterophyllum*, *Dictyophyllum*, *Pagiophyllum*, *Baiera guilhaumatii*, and *Podozamites lanceolatus*. They also changed the 'Tsungshan Formation' to 'Xiangshan Layers', and regarded its age as being the Early Jurassic. *Sze and Chow (1962)* used the term 'Xiangshan Group' for the previous 'Xiangshan Layers', and this has been the convention followed ever since. The age of the Xiangshan Group has been concluded by various authors to range from the Late Triassic to the Middle Jurassic. *Ju (1987)* divided the Xiangshan Group into a lower South Xiangshan Formation and an upper North Xiangshan Formation, respectively. The standard section of the lower part of the Xiangshan Group (the South Xiangshan Formation, the Lower Jurassic) is 394 meters thick in South Xiangshan in Nanjing, and has yielded abundant fossil plants. The standard section of the upper part of the Xiangshan Group (the North Xiangshan Formation, the Middle Jurassic) is in North Xiangshan in Nanjing. It is 1005 meters thick, and has only yielded a few stem fossils (*Ju, 1987*). Based on fossil plants, *Cao (1982)* thought that the age of the South Xiangshan Formation could not be later than the Early Jurassic, and considering that the early Early Jurassic flora (in the middle and lower parts of the Guanyintan Formation in southwest Hunan) (*Zhou and Li, 1980*) is biostratigraphically below the South Xiangshan Formation, *Cao (1982)* regarded the age of the lower part of Xiangshan Group (=South Xiangshan Formation) as the middle-late Early Jurassic.

The South Xiangshan Formation (lower part of the Xiangshan Group) has yielded abundant bivalve and plant fossils. Its outcrops are scattered in Jiangning, Longtan, and Zhenjiang (all in the suburbs of Nanjing). In these areas, the outcrops are well exposed and especially fossiliferous near the South Xiangshan and Cangbomen regions. The formation includes sandstones, siltstones, shales, carbonaceous shales, and coal seams. There are abundant plant fossils in the South Xiangshan Formation, and almost all of the plants in the Xiangshan Group are from this formation. Various authors have collected fossil plants of the Xiangshan Flora (*Cao, 1982*; *Cao, 1998*; *Cao, 2000*; *Wang et al., 1982*; *Huang, 1983*; *Huang, 1988*; *Ju, 1987*). According to *Cao (1982)*, *Cao (1998)*, *Cao (2000)*, *Wang et al. (1982)*, and *Ju (1987)*, the Xiangshan Group includes at least 46 genera of plants and is very similar to the flora of the Hsiangchi Group in western Hubei. Cycadophytes (34%) dominate the flora, and ferns are the second most dominant group (20%), among which Dipteridaceae play an important role. Ginkgoales are also abundant (19%) (*Ju, 1987*). The important and frequently observed taxa include *Hysterites*, *Selaginellites*, *Equisetites* cf. *lateralis*, *E.* aff. *multidentatus* Oishi, *E. sarrani* (Zeiller) Halle, *Neocalamites hoerensis* (Schimper) Halle, *N. dangyangensis* Chen, *Marattiopsis asiatica* Kawasaki, *M. hoerensis* (Schimper) Schimper, *Todites goeppertianus* (Münster) Krasser, *T. princeps* (Presl) Gothan, *Osmundopsis* Harris, *Cladophlebis denticulata* (Brongniart) Fontaine, *C. goeppertianus* (Münster) Krasser, *C. raciborskii* Zeiller, *Spiropteris* Schimper, *Phlebopteris polypodioides* Brongniart, *Danaeopsis* Heer ex Schimper, *Thaumatopteris pusilla* (Nathorst) Oishi et Yamasita, *Dictyophyllum nathorstii* Zeiller, *D. nilssonii* (Brongniart) Goeppert, *Clathropteris meniscioides* Brongniart, *Cl. platyphylla* Goeppert, *Cl. obovata* Oishi, *Coniopteris hymenophylloides* (Brongniart) Seward, *Thinnfeldia* Ettingshausen, *Augustiphyllum yaobuensis* Huang, *Scoresbya dentata* Harris, *Pterophyllum firmifolium* Ye, *Pt. propinquum* Goeppert, *Pt. subaequale* Hartz, *Nilssonia complicatis* Li, *N. orientalis* Heer, *N. minor* Harris, *N.* cf. *compta* (Schenk) Ye, *N.* cf. *polymorpha* Schenk, *N. pterophylloides* Nathorst, *N.* cf. *saighanensis* Seward, *N. taeniopterioides* Halle, *N. parabrevis* Huang, *N. moshanensis* Huang, *Nilssoniopteris vittata* (Brongn.) Florin, *Ctenis* Lindley et Hutton, *Ctenozamites*

cf. *ptilozamioides* Zhou, *C.* cf. *cycadea* (Berger) Schenk, *Cycadolepis corrugata* Zeiller, *Anomoza-mites* cf. *minor* Nathorst, *A.* cf. *major* (Brong) Huang, *A.* cf. *inconstans* (Goeppert) Schimper, *A. quadratus* Cao, *Tyrmia nathorstii* (Schenk) Yeh, *T. latior* Ye, *T. lepida* Huang, *T. susongensis* Cao, *Otozamites minor* Tsao, *Ot. hsiangchiensis* Sze, *Ot. mixomorphus* Ye, *Ot. tangyanensis* Sze, *Ptilo-phyllum hsingshanense* (Wu) Cao, *Pt. contiguum* Sze, *Pt. pecten* (Philips) Morris, *Hsiangchiphyllum trinervis* Sze, *Ginkgoites* cf. *tasiakouensis* Wu et Li, *G.* cf. *sibiricus* (Heer) Seward, *G.* cf. *magnifolius* Du Tiot, *Baiera* cf. *furcata* (L. et H.) Braun, *B. asadai* Yabe et Oishi, *B. guilhaumatii* Zeiller, *B. multi-partita* Sze et Lee, *B.* cf. *gracilis* Bunbury, *Sphenobaiera huangii* (Sze) Hsu ex Li, *S. spectabilis* (Nath.) Florin, *Czekanowskia rigida* Heer, *C. hartzii* Harris, *Phoenicopsis* Heer, *Ginkgodium* Yokoyama, *Des-miophyllum* Lesquereux, *Stenorachis* (Nathorst) Saporta, *Vittifoliolum multinerve* Zhou, *Pityophyllum longifolium* (Nathorst) Möller, *Podozamites lanceolatus* (L. et H.) Braun, *Ferganiella* Prynada, *Elato-cladus* Halle, *Swedenborgia cryptomerioides* Nathorst, *Taeniopteris* cf. *richthofenii* (Schenk) Sze, *T. inouyei* Tateiwa, *Conites* and *Carpolithus* (*Figure 3—figure supplement 1*; *Figure 4—figure supple-ment 1*).

## Palynological assemblage

Preliminary analysis of the strata yielding *Nanjinganthus* has recognized abundant palynomorphs. The palynoflora includes *Anapiculatisporites* sp., *Annulispora folliculosa* (Rogalska) De Jersey, *Con-tignisporites* sp., *Cyathidites australis* Couper, *C. minor* Couper, *Deltoidospora* sp., *D. minor* Pocock, *Dictyophyllidites harrisii* Couper, *D. mortonii* (De Jersey) Playford and Dettmann, *Gleicheniidites* sp., *G. senonicus* Ross, *Ischyosporites* sp., *I. variegatus* (Couper) Schultz, *Leptolepidites verrucatus* Couper, *Manumia delcourtii* (Pocock) Dybkjær, *Neoraistrickia ramosus* (Balme and Hennelly) Hart, *Osmundacidites wellmanii* Couper, *Polycingulatisporites triangularis* (Bolchovitina) Playford and Dett-mann, *Punctatosporites* sp., *Retitriletes austroclavatidites* (Cookson) Döring et al., *R. clavatoides (Couper)* Döring et al., *Sestrosporites pseudoalveolatus* (Couper) Dettmann, *Striatella scanica* (Nils-son) Filatoff and Price, *S. seebergensis* Mädler, *Alisporites* sp., *A. robustus* Nilsson, *Callialasporites dampieri* (Balme) Dev, *C. minus* (Tralau) Guy, *C. trilobatus* (Balme) Dev, *C. turbatus* (Balme) Schulz, *Cerebropollenites macroverrucosus* (Thiergart) Pocock, *Chasmatosporites* sp., *C. apertus* (Rogalska) Nilsson, *C. hians* Nilsson, *Classopollis chateaunovi* Reyre, *C. classoides* (Pflug) Pocock and Jansonius, *C. meyeriana* (Klaus) De Jersey, *C. simplex* (Danzé-Corsin and Laveine) Reiser and Williams, *Cycado-pites* sp., *C. follicularis* Wilson and Webster, *Ephedripites* sp., *Monosulcites* sp., *M. minimus* Cook-son, *Perinopollenites elatoides* Couper, *Platysaccus* sp., *Podocarpidites* sp., *Quadraeculina anellaeformis* Maljavkina, *Q. enigmata* (Couper) Xu and Zhang, *Q. minor* (Pocock) Xu and Zhang, *Spheripollenites psilatus* Couper, *Vitreisporites pallidus* Nilsson (*Figure 2—figure supplement 1*) (*Santos et al., 2018*). This palynological assemblage suggests a latest Early Jurassic age for *Nanjinganthus*.

## Isotopic dating

The samples were processed by crushing, initial heavy liquid and subsequent magnetic separation at Langfang Yuneng Rock Mineral Separation Technology Service Co., Ltd. in Langfang City. More than 1000 grains of zircons were hand-picked under a binocular microscope. More than 200 grains of rep-resentative zircons for each sample were coined in epoxy resin mounts, ground and polished to expose the central part of zircons, and then photographed under microscope in transmitting light and reflected light. Afterward, the internal structure of the zircons was studied by means of cathodo-luminescence (CL) imaging at the Beijing Gaonianlinghang Technology Co., Ltd. in Beijing City. U-Pb dating of these samples were carried out using laser ablation multicollector inductively coupled plasma mass spectrometry (LA-MC-ICP-MS) at the Tianjin Institute of Geology and Mineral Resour-ces. The laser beam was 35 μm in diameter. Concentrations of U, Th, and Pb elements were cali-brated using SRM 610 as the external reference standard. For the analysis method please see refer *Li et al., 2009*. Repeated analyses of standards yielded precisions at better than 10% for most elements. $^{207}Pb/^{206}Pb$, $^{206}Pb/^{238}U$, $^{207}Pb/^{235}U$ and $^{208}Pb/^{232}Th$ ratios and apparent ages were calcu-lated using ICPMSDataCal software (*Liu et al., 2010a*; *Liu et al., 2010b*) and corrected for both instrumental mass bias and depth dependent elemental and isotopic fractionation using zircon GJ-1 as the external standard. U-Pb age Concordia diagram and histograms apparent ages diagram were drawn by using ISOPLOT (ver.3) (*Ludwig and Ludwig, 2003*).

There was no previous isotopic age for the Xiangshan Group. We sampled the layers above the fossiliferous layers (*Figure 1—figure supplement 1*) and picked zircon grains for U/Pb dating. The zircon grains appeared to be reworked (*Supplementary file 2*), with ages ranging from 2738 Ma to 207 Ma (67 zircon grains with the concordance >90% from 168 zircon grains), and 207 Ma (two zircon grains) is the youngest age (*Figure 1—figure supplement 1*). Most of the zircon grains were of magmatic origin with internal oscillation belts and high Th/U values, implying a granitic provenance. So the upper limit age of *Nanjinganthus* is 207 Ma (the Late Triassic).

Taking all dating information into consideration, we think that the age of *Nanjinganthus* falls in the scope ranging from 174 to 207 Ma and is closer to 174 Ma (the latest Early Jurassic). Such a conclusion on absolute age of *Nanjinganthus* is in agreement with megafossil biostratigraphical analysis (*Cao, 1982*; *Huang, 1983*; *Huang, 1988*; *Ju, 1987*), although *Neocalamites horridus* was previously known only in the Late Triassic (*Zan et al., 2012*).

## Materials

The fossils studied here were collected from an outcrop of the South Xiangshan Formation at a quarry owned by the Xiaoyetian Cement Company Ltd. in the northeastern suburb of Nanjing, Jiangsu, China (N32°08′ 19″, E118°58′ 20″) (*Figure 1—figure supplement 1*). Plant fossils of the formation have been extensively studied by various scholars (*Sze and Chow, 1962*; *Cao, 1982*; *Cao, 1998*; *Cao, 2000*; *Wang et al., 1982*; *Huang, 1983*; *Huang, 1988*; *Ju, 1987*), and our collection from the local outcrop indicates that the fossil plants closely associated with *Nanjinganthus* constitute a flora dominated by Dipteridaceae (*Clathropteris*) and various cycadophytes (mainly *Nilssonia*, *Ptilophyllum*, and *Pterophyllum*), which is consistent with previous works. Some of these associated plant fossils are shown in *Figure 3—figure supplement 1* and *Figure 4—figure supplement 1*.

## Methods

The specimens were initially photographed using a Sony ILCE-7 digital camera. The sediment covering the specimens was dégaged using a JUN-AIR pneumatic drill, and the details of the fossils were observed and photographed using a Nikon SMZ1500 stereomicroscope equipped with a Digital Sight DS-Fi1 camera. Organically preserved sepals and petals were processed with 40% peroxide for cuticle analysis according to routine methods, and the processed cuticles and cleaned organic material of the sepals and petals were observed and photographed using the Rhod fluorescent light in a Zeiss Z2 Imager with an AxioCam HRc camera. Extended-focus images were generated using the Z-stack function in an AxioVs40 × 64 V4.9.1.0. The removed cuticles were coated with gold and observed using a Leo 1530 VP scanning electron microscope (SEM), and serial pictures were obtained after the internal details of the flower were exposed through grinding with a pneumatic drill. One of the organically-preserved petals was embedded in resin and sectioned for light microscopy and transmission electron microscopy (TEM). One fragment of the distal portion of a flower embedded in sediments was observed by Micro Computed Laminography (Micro-CL) (*Wei et al., 2017*) to show the dendroid style embedded in the sediments. All photographs were saved in TIFF format and assembled for publication using Photoshop 7.0.

## Acknowledgements

We thank Ms Huijun Ma, Brandon Y. Wang, and Mingzhi Fu for their help collecting the valuable specimens, Ms. Chunzhao Wang for help with SEM during this research, and Dr. Walter Judd at the University of Florida for comments and suggestions. This research was supported by the Strategic Priority Research Program of Chinese Academy of Sciences (Grant No. XDB26000000), the National Natural Science Foundation of China (41688103, 91514302, 41572046) awarded to XW. This is a contribution to UNESCO IGCP632. We declare no competing interests. We appreciate the kind and constructive helps from Dr. David Taylor, Hongqi Li, Ian Baldwin and an anonymous reviewer, who made our manuscript much improved.

## Additional information

### Funding

| Funder | Grant reference number | Author |
|---|---|---|
| National Natural Science Foundation of China | 41688103 | Xin Wang |
| Chinese Academy of Sciences | XDPB26000000 | Xin Wang |
| National Natural Science Foundation of China | 91514302 | Xin Wang |
| National Natural Science Foundation of China | 41572046 | Xin Wang |

The funders had no role in study design, data collection and interpretation, or the decision to submit the work for publication.

### Author contributions

Qiang Fu, Conceptualization, Resources, Writing—review and editing; Jose Bienvenido Diez, Manuel García Ávila, Hang Chu, Yemao Hou, Pengfei Yin, Kaihe Du, Data curation, Writing—review and editing; Mike Pole, Zhong-Jian Liu, Guo-Qiang Zhang, Writing—review and editing; Xin Wang, Conceptualization, Resources, Writing—original draft

### Author ORCIDs

Qiang Fu (iD) https://orcid.org/0000-0002-6948-3747
Jose Bienvenido Diez (iD) http://orcid.org/0000-0001-5739-7270
Pengfei Yin (iD) http://orcid.org/0000-0002-4520-5347
Xin Wang (iD) http://orcid.org/0000-0002-4053-5515

### Decision letter and Author response

Decision letter https://doi.org/10.7554/eLife.38827.024
Author response https://doi.org/10.7554/eLife.38827.025

## Additional files

### Supplementary files

• Supplementary file 1. Number of flowers on each *Nanjinganthus* specimen.
DOI: https://doi.org/10.7554/eLife.38827.020

• Supplementary file 2. Summary of zircon LA–ICP–MS U–Pb data for fossiliferous layers samples of the South Xiangshan Formation. Online Information 3D virtual image of the holotype of *Nanjinganthus*. Click on the link, using ctrl/shift and mouse, you can manipulate the image for your observation.
DOI: https://doi.org/10.7554/eLife.38827.021

• Transparent reporting form
DOI: https://doi.org/10.7554/eLife.38827.022

### Data availability

All data generated or analysed during this study are included in the manuscript and supporting files. Source data files have been provided as Supplementary files 1 and 2, Figure 1—figure supplement 1, Figure 2—figure supplement 1, Figure 3–figure supplement 1, Figure 4—figure supplement 1 and Figure 5—figure supplement 1.

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
