## [Decision Letter]

[Editors’ note: a previous version of this study was rejected after peer review, but the authors submitted for reconsideration. The first decision letter after peer review is shown below.]

Thank you for submitting your work entitled "*Nanjinganthus*: An Unexpected Flower from the Jurassic of China" for consideration by *eLife*. Your article has been reviewed by three peer reviewers, including David Taylor as the Reviewing Editor and Reviewer #1, and the evaluation has been overseen by a Senior Editor. The following individual involved in review of your submission has agreed to reveal their identity: Hongqi Li (Reviewer #2). A further reviewer remains anonymous.

The decision to reject this manuscript was difficult. The material is unique and provides a huge amount of data not normally available. Yet justification of such a new taxon requires a level of description that not only shows the characters, but also excludes the possibility of relationships to living and fossil seed plant groups.

Below is a summary of problems based both on the reviews and discussions between the reviewers. We hope that they will form the basis of a revision that will allow the work to be improved and accepted for its value. Please note that additional details are discussed in the reviews.

1) Diagnosis should not include all of the description and concentrate on the unique, unequivocal characters. For example words such as "probably", "more or less" and "possible" do not have a place in a diagnosis.

2) There is not enough discussion of each organ explaining why it is as named and not something else. You do discuss but do not reference figures again.

3) Cross referencing of figures to other figures (Figure 2I is same as 3A-C).

4) Need specimen numbers so can easily refer to each other.

5) Is the carpel closed? The evidence of closed ovary, explanation why those appearing open are to be excluded, such as due to preservation. Must discuss those case figures where it looks open. The best demonstration of closed is the Figure 7A-C series, showing organic connection of "dendroid style". Figure 2H also appears to show it. 3I does say that it is broken. What about 4C and 5H?

6) What is the branched "dendroid style" structure? Must exclude other possibilities. Also, any cases of flowers with "dendroid style" also with seeds? E.g., "dendroid style" is quite similar to a mature pollen cone of Taxus (no mention of this in the manuscript). Need more on description. You only mention Figure 4 specimen with the "dendroid style": Flower 1 in Figure 1D shown in Figure 2H; Flower 2 in Figure 1D shown in Figure 2I, Figure 3A-C; Flower 10 in Figure 1E shown in 5GF, 6A-C; PB22282 shown in Figure 7A-B. It is weird, but based on those flowers, it is always centrally placed, has a naked portion at base, it projects past the bracteoid structures and connects in the same line, and first branches opposite to subopposite. It does not seem to project below the line between the attachment of the last bracteiod structures "petals". Could they be germinating seeds?

7) Are the sterile organs bracts/scales or perianth organs? Are the "ribs" really veins, and are they clearly different in each organ? Also, these seem to have a 2/5 phylotaxy, not a simple spiral.

8) Stamen details are need. Since only one found, and it does not have an ovary, probably should not assume bisexual.

9) Other evidence of expected angiosperm characters, e.g., pollen size and ornamentation, could strengthen case. These pollen grains are the expected size and ornamentation (based on Doyle's work and separately my work). Would be great to have ultrastructure which IS unique to angiosperms.

10) There are no good comparisons among these fossils and taxa that easily conforms with these fossils. If not angiosperm, then what? New group?

*Reviewer #1:*

There have been many predications of Jurassic angiosperms. Although there have been several claims in recent years of Jurassic angiosperms, the totality of evidence was unconvincing. In my opinion, there are three major criteria for the evidence to be compelling:

1) The fossil has as multiple angiosperm characters. If not then the additional characters must not be defining characteristics of other groups of living or fossil seed plants.

2) Age and dating of the sediment the fossil species is found is reliable, including that the fossils were collected by the authors so the placement and stratigraphy is reliable.

3) The characteristics are consistent with hypothesis of what the ancestral character states might be. These can by phylogenetic (synapomorphies) or conceptual.

My review has been to carefully work through the figures and captions. I compliment the authors on a detailed captions and descriptions, and excellent use of arrows to describe what they see on the plates. There is no doubt of what the authors believe they are seeing, even if the reader disagrees with the interpretation. The biggest strength is the larger number of specimens with different types of preservation and the multiple views of the same organs.

The abstract is strong. The description is also fine, although there a couple of mistakes where the referred figures are the wrong ones.

The following is my evaluation of the three criteria.

1) Yes, including closed ovary, stigma and style, 4-parted anther, bisexual reproductive organ, anther with filament, alternate/spiral phylotaxy, low ovule/seed number, etc. They are correct that there is no other living of fossil group that has this suite of characters.

2) The biostratigraphy looks strong with both macrofossil and microfossil evidence. But they should explicitly say whether the megafossil evidence supports the age. The Ar40/Ar39 is nice but not overly conclusive.

3) This is the weakest part, but sufficient. Carpel definition is perfunctory and lacking references.

In total, I believe the authors can correctly claim they have described an early angiosperm from the Jurassic.

This paper should be published after revisions.

*Reviewer #2:*

Certainly, this is a very interesting fossil plant as it could provide significant morphological data to help to solve the century mystery, Origin of Angiosperms. However, I do have following five major concerns:

1) The authors interpret the fossil as angiosperms because the "ovarian roof has no any opening" and contained "seeds" inside. However, Figures 6F 4A, 4D and I, 5C show "seeds" exposed clearly in the center, so are the openings natural (that means the "receptacle/ovary" is never closed) or the openings are the broken rims of the style bases in the center of ovarian roof. If it is the former, the plant should be a gymnosperm. If it is the latter case, the nature of the "style" should be further examined. In Figure 6B in the center of the "ovarian roof", could the two obviously arching structures be the "style" base that extended upwards into the hollow lower part of "style" as showing in Figure 6A? If so, then could the "style" be a tubular structure extending outwards, like the protruding micropyle of gnetophytes? Since neither dendroid style is found in living angiosperms, nor a dendroid micropyle is found in gnetophytes, more detailed work is needed to reveal its nature. Suggestion: make serial sections on some specimen (e.g., the one in Figure 2H) to reveal the real/3D connection between the "dendroid style" and the "ovarian roof."

2) Figure 2F shows another opening with two "seeds" exposed in the edge in the top area, and this opening seems to be natural too. Could it suggest the plant of gymnosperms? Authors really need to demonstrate the nature of this opening. Also interestingly, the two exposed "seeds" in Figure 6H exhibit different colored zones. Are they real seeds or leaf modifications with veins, such as bracteoles? Figure 6G shows a seed with two horizontal groves. How to interpret the "groves"? Suggestion: apply fluorescent microscopy to examine the "seeds", to see if new evidence can be found.

3) The most bizarre structure is the "dendroid style". How many "flowers" have it? If it is a style, the end of each branch should be a stigma. Even if pollen germinated pollen tubes, the pollen wall could still be attached onto the stigmas. Suggestion: use fluorescence microscope and SEM to scan over those branch ends to make sure they are real fossil stigma with pollen.

4) The "stamen" presented in Figure 7J-K is not so convincing so that even the authors call it "possible stamen". Suggestion: make serial sections of specimens of flower buds to figure out the real shape of the stamen and find out more *in situ* pollen. It will be very significant if the pollen type matches up with those found on the "stigma".

5) The "bracts", "scales", "sepals", and "petals" appear all similar (Figure 5I), including cellular structures (Figure 10), except for size. Even the authors use "b" for a scale in Figure 2I. Could they be all bracts and bracteoles like those on the cones of gnetophytes? How to distinguish them from each other?

*Reviewer #3:*

The principal goal of this manuscript is to describe and present a new fossil taxon *Nanjinganthus dendrostyla*, an angiosperm from Early Jurassic South Xiangshan Fm. that outcrops in China. This new proposed taxon is based on "284 individual flowers on 28 slabs preserved in various orientations and states". They specimens are impressions/compressions. I have major concerns about the interpretation of these fossils and the inclusions of such interpretations within the diagnosis.

Unfortunately, I have to disagree with the opinion of the authors that the fossils are well-preserved; the fossils are indeed poorly preserved (the authors even need to add lines on the fossil photos to show "organs" and characters). They placed this proposed taxon within the flowering plants based on their interpretation of structures that the authors claim they are observing. This is clear in the diagnosis (which it is basically a description and not a diagnosis). The diagnosis, description, treatment, Discussion and conclusions are not consistent with the evidence presented in the figures. These fossils can easily be conifer shoots preserved in different positions (the comparisons and Table 1 presented in the subsection “Eliminating Alternative Interpretations” are extremely vague) for example the cuticular characters of the "petals" are typical of conifers.

[Editors’ note: what now follows is the decision letter after the authors submitted for further consideration.]

Thank you for submitting your article "*Nanjinganthus*: An Unexpected Flower from the Jurassic of China" for consideration by *eLife*. Your article has been reviewed by two peer reviewers, including David Taylor as the Reviewing Editor and Reviewer #1, and the evaluation has been overseen by Ian Baldwin as the Senior Editor. The following individual involved in review of your submission has also agreed to reveal their identity: Hongqi Li (Reviewer #2).

The reviewers have discussed the reviews with one another and the Reviewing Editor has drafted this decision to help you prepare a revised submission.

After discussion between the reviewers we do have two more important suggestions. We decided it would be good to keep the description of the unattached potential stamen, but we strongly suggest that in the text and the legend for Figure 3GH it explicitly explains the relationship of the organ and the associated flower (that they are *not* attached). In addition, the holotype is not a rock block but an individual fossil. You need to pick one fossil with sufficient detail as the type specimen.

These editorial suggestions are meant to help improve the scientific impact by increasing the scientific accuracy of your text, but also to help you find a more neutral tone with which to present these revolutionary findings. Your work will surely stir up the community, and it's best to remain as "evidence based" as possible in the presentation of your work.

*Reviewer #1:*

I have had the opportunity to review 4 versions of this manuscript. This version is the strongest of all. In my opinion, the authors have fulfilled the three major criteria for the evidence on a flower to be compelling:

1) The fossil has as multiple angiosperm characters. If not then the additional characters must not be defining characteristics of other groups of living or fossil seed plants.

2) Age and dating of the sediment the fossil species is found is reliable, including that the fossils were collected by the authors so the placement and stratigraphy is reliable.

3) The characteristics are consistent with hypothesis of what the ancestral character states might be. These can by phylogenetic (synapomorphies) or conceptual.

Whether or not an angiosperm, this is a unique fossil species and should be published. The strengths are the figures, captions and descriptions. This discussion of relationship to other Jurassic plants is good, as in general the discussion of the organs. I would publish this with very minor corrections.

Need corrections:

Zeng et al., 2014 is not referenced.

Holotype. The holotype is not a rock block but of an individual fossil. You need to pick one fossil with sufficient detail as the type specimen.

*Reviewer #2:*

I am very pleased to find that this version of the manuscript is in a much better level now, and I would recommend it for publication, with the following aspects to be changed or addressed:

1) Insert "Noncarpellate Epigynous" between "Unexpected" and "Flower" in the title, because noncarpellate flower is more significant than its occurrence in Jurassic in my opinion, and epigynous flower is also very important. So, the "Unexpected" refers to the three aspects, and will attract more attention from not only paleobotanists but also modern botanists.

2) Therefore, "Noncarpellate Epigynous" should be emphasized more in the Discussion! For the part "This discrepancy suggests *either*.…. reassessed", I would recommend to adopt the style of Sattler and Lacrois's statement, "With regard to the evolution of basal placentation in Basella and other taxa of Angiosperms three possibilities exist: 1) It is derived from the carpellate condition, 2) It is primitive and the carpellate condition is derived, 3) Both carpellate and noncarpellate organizations have coexisted during the evolution of Angiosperms which may have been monophyletic or polyphyletic". And *Nanjinganthus* supports their second possibility! The three aspects and their implications should also be mentioned again in the Conclusion.

3) Since the "possible stamen" is not connected to any flower, please remove any related part *thoroughly*, even from "remarks", Figures, and the reconstruction drawing. Introducing this un-connected "possible stamen" will only weaken the credibility of whole paper!

[Editors' note: further revisions were requested prior to acceptance, as described below.]

Thank you for resubmitting your work entitled "An Unexpected Noncarpellate Epigynous Flower from the Jurassic of China" for further consideration at *eLife*. Your revised article has been favorably evaluated by Ian Baldwin (Senior Editor), and a Reviewing Editor.

The manuscript has been improved but there are some remaining issues that need to be addressed before acceptance, as outlined below:

1) That the authors temper their conclusion that their data demonstrates that this is an early Angiosperm flower.

2) That the authors make sure the U-Pb analyses are explained in Materials and methods, and the results are properly presented.

3) An important source for confirming the age comes from the pollen assemblage. But there is no reference to a comparison from the literature. That the authors provide references and a brief discussion how your assemblage is similar, or a temporal range chart of the palynomorphs and a brief discussion why it supports the age.

---

## [Author Response]

[Editors’ note: the author responses to the first round of peer review follow.]

The decision to reject this manuscript was difficult. The material is unique and provides a huge amount of data not normally available. Yet justification of such a new taxon requires a level of description that not only shows the characters, but also excludes the possibility of relationships to living and fossil seed plant groups.

Here, we document a Jurassic angiosperm flower based on more than two hundred specimens of flowers preserved in various states from the Early Jurassic of Nanjing, Jiangsu, China. For the first time, there are abundant enough Jurassic flowers with features characteristic of angiosperm. This discovery refutes a 57-year-old doctrine, topples the current belief about evolutionary history and systematics of angiosperm, because this fossil significantly extends the history of angiosperms. This version is the result of modification after the suggestions from the editorial office and experts. The major changes include our addressing the concerns from the reviewers, and we have provided detailed point-to-point answers to their questions. In addition, more detailed supporting information is provided in our supplementary material.

Below is a summary of problems based both on the reviews and discussions between the reviewers. We hope that they will form the basis of a revision that will allow the work to be improved and accepted for its value. Please note that additional details are discussed in the reviews.1) Diagnosis should not include all of the description and concentrate on the unique, unequivocal characters. For example words such as "probably", "more or less" and "possible" do not have a place in a diagnosis.

Thanks. We have corrected these mistakes.

2) There is not enough discussion of each organ explaining why it is as named and not something else. You do discuss but do not reference figures again.

We have provided the definitions and demarcations for the terms we used, and marked them in the reconstruction.

3) Cross referencing of figures to other figures (Figure 2I is same as 3A-C).

We have adjusted the figures and cross-referred each other.

4) Need specimen numbers so can easily refer to each other.

We have added this information for every figure.

5) Is the carpel closed?

We have not seen typical carpels in classical sense in the specimens, so we do not know whether there are carpels in the classical sense in *Nanjinganthus*. So we cannot answer this question.

The evidence of closed ovary, explanation why those appearing open are to be excluded, such as due to preservation. Must discuss those case figures where it looks open.

We have added discussion addressing this concern. Intact ovarian roof is seen only in Figures 4C, 5H (probably plus Figures 7E-G). The open-appearing ovary is due to preservation.

The best demonstration of closed is the Figure 7A-C series, showing organic connection of "dendroid style". Figure 2H also appears to show it. 3I does say that it is broken.

3I is partially broken.

What about 4C and 5H?

This is intact, so no ovules/seeds are visible because they are eclipsed.

6) What is the branched "dendroid style" structure? Must exclude other possibilities.

We have addressed this concern. The appearance makes it look like some male cones in conifers, but this possibility has been eliminated in our Discussion, due to lacking pollen grains (which are more likely than all other parts to be preserved in fossils) and penetrating cone axis.

Also, any cases of flowers with "dendroid style" also with seeds?

So far not yet.

E.g., "dendroid style" is quite similar to a mature pollen cone of Taxus (no mention of this in the manuscript).

We have discussed and eliminated this possibility.

Need more on description. You only mention Figure 4 specimen with the "dendroid style": Flower 1 in Figure 1D shown in Figure 2H; Flower 2 in Figure 1D shown in Figure 2I, Figure 3A-C; Flower 10 in Figure 1E shown in 5GF, 6A-C; PB22282 shown in Figure 7A-B.

It has been seen on 4 slabs for 10 times.

It is weird, but based on those flowers, it is always centrally placed, has a naked portion at base, it projects past the bracteoid structures and connects in the same line, and first branches opposite to subopposite. It does not seem to project below the line between the attachment of the last bracteiod structures "petals". Could they be germinating seeds?

Taking all information into consideration, we think the dendroid style is a structure that is inserted on the center of the ovarian roof and has little to do with the seeds inside ovary.

7) Are the sterile organs bracts/scales or perianth organs?

We think this is possible. There might be a transition from one category to another. Their difference is likely hinged with their position in the flower.

Are the "ribs" really veins, and are they clearly different in each organ?

We think they are veins, judged basing on transmission light microscopy and stereomicroscopy. They are not obvious in bracts and scales.

Also, these seem to have a 2/5 phylotaxy, not a simple spiral.

Our observation does not allow us to make any statement about this.

8) Stamen details are need. Since only one found, and it does not have an ovary, probably should not assume bisexual.

We have accepted this suggestion and are not talking about possible bisexuality in the newer version. The stamen is also removed from the reconstruction.

9) Other evidence of expected angiosperm characters, e.g., pollen size and ornamentation, could strengthen case. These pollen grains are the expected size and ornamentation (based on Doyle's work and separately my work). Would be great to have ultrastructure which IS unique to angiosperms.

The pollen and stamen are rare, tentative, sparse evidence now. We wish to refrain us from making bold statements.

10) There are no good comparisons among these fossils and taxa that easily conforms with these fossils. If not angiosperm, then what? New group?

We wish there are. But current situation does not allow us to do so. Considering the early Jurassic age, it would be very suspicious if it could be directly related to any known angiosperms. The possibility of a gymnosperm with enclosed ovules has been a candidate in our mind. However, according to the current consensus, there is no gymnosperm with enclosed ovules yet (although there are indeed seeds protected in various ways).

Reviewer #1:

There have been many predications of Jurassic angiosperms. Although there have been several claims in recent years of Jurassic angiosperms, the totality of evidence was unconvincing. In my opinion, there are three major criteria for the evidence to be compelling:1) The fossil has as multiple angiosperm characters. If not then the additional characters must not be defining characteristics of other groups of living or fossil seed plants.

Thanks. We have modified our manuscript with this in mind.

2) Age and dating of the sediment the fossil species is found is reliable, including that the fossils were collected by the authors so the placement and stratigraphy is reliable.

Thanks. The strata have been investigated for a long time. So far there is no controversy on the age yet. Various evidence seems to agree on the Early Jurassic age.

3) The characteristics are consistent with hypothesis of what the ancestral character states might be. These can by phylogenetic (synapomorphies) or conceptual.

We cannot fully agree with this, although we think this is a rational expectation. Before talking about “consistent with hypothesis”, the first question is “which hypothesis”? And why that theory? All currently available hypotheses are not justified enough to be such an ideal hypothesis yet. The second question is “Is this hypothesis reliable, believable, supported by fossil evidence?” In our eyes, there is none. It is completely possible that the hypothesis is simply taken for granted, based on no fossil evidence. Our fossils do not fit in any seek-images of ancestral angiosperms. This is not our fault or our fossils’ fault. It is the fault of the so-called “theories”, which do not reflect the reality. This is the part of botany that should be changed, not our fossils. Our fossils are not the only case refuting the “theories”, for example, *Archaefructus* from the Early Cretaceous does not look like *Magnolia* or *Amborella* at all. So please forgive our fossils from not fitting in the “theoretical expectation”. We believe, if the theory is correct, our fossils will definitely fit in its expectation. *Nanjinganthus* falls within the scope of the expectation of the Unifying Theory advanced by Wang (2010, 2018).

My review has been to carefully work through the figures and captions. I compliment the authors on a detailed captions and descriptions, and excellent use of arrows to describe what they see on the plates. There is no doubt of what the authors believe they are seeing, even if the reader disagrees with the interpretation. The biggest strength is the larger number of specimens with different types of preservation and the multiple views of the same organs.The abstract is strong. The description is also fine, although there a couple of mistakes where the referred figures are the wrong ones.

Thanks. We have double checked the figure referring in the newer version.

The following is my evaluation of the three criteria.1) Yes, including closed ovary, stigma and style, 4-parted anther, bisexual reproductive organ, anther with filament, alternate/spiral phylotaxy, low ovule/seed number, etc. They are correct that there is no other living of fossil group that has this suite of characters.

Thanks.

2) The biostratigraphy looks strong with both macrofossil and microfossil evidence. But they should explicitly say whether the megafossil evidence supports the age. The Ar40/Ar39 is nice but not overly conclusive.

Thanks. We have added this statement.

3) This is the weakest part, but sufficient. Carpel definition is perfunctory and lacking references.

The carpel, at least in the classical sense, is not a term that can be applied universally in angiosperms. Some botanists have refuted the term of “carpel”, although we do not. However, we have given some references about this in the newer version.

Reviewer #2:

Certainly, this is a very interesting fossil plant as it could provide significant morphological data to help to solve the century mystery, Origin of Angiosperms. However, I do have following five major concerns:1) The authors interpret the fossil as angiosperms because the "ovarian roof has no any opening" and contained "seeds" inside. However, Figures 6F 4A, 4D and I, 5C show "seeds" exposed clearly in the center, so are the openings natural (that means the "receptacle/ovary" is never closed) or the openings are the broken rims of the style bases in the center of ovarian roof.

This breaking is not original, not natural. It is rather preservation artifacts. The intact ovarian roof is seen only in Figures 4C and 5H, in which case no ovules or seeds are seen, implying the ovules and seeds are fully enclosed. Style base and ovarian roof are the same thing, because the style is located on the top of the ovarian roof. It is impossible to distinguish these two from each other in *Nanjinganthus*.

If it is the former, the plant should be a gymnosperm. If it is the latter case, the nature of the "style" should be further examined. In Figure 6B in the center of the "ovarian roof", could the two obviously arching structures be the "style" base that extended upwards into the hollow lower part of "style" as showing in Figure 6A? If so, then could the "style" be a tubular structure extending outwards, like the protruding micropyle of gnetophytes?

This alternative does not exist. If this is correct, then the logical inference is that all organs including scales, sepals, petals are borne on the style. This is never seen in any plant including Gnetales.

Since neither dendroid style is found in living angiosperms, nor a dendroid micropyle is found in gnetophytes, more detailed work is needed to reveal its nature.

Dendroid (branched) style has been seen in Poaceae and Malvaceae. However, it is premature to correlate *Nanjinganthus* directly with Poaceae and Malvaceae based on this single feature.

Suggestion: make serial sections on some specimen (e.g., the one in Figure 2H) to reveal the real/3D connection between the "dendroid style" and the "ovarian roof."

Thanks for the suggestion. Current technology does not allow us to do serial sectioning. We have removed some material from Figure 2H for SEM observation, through which the stoma in Figure 8H is seen, and we did not see any possibility of 3D configuration of the ovary to be visualized. However, the connection between the style and ovarian roof have been clearly seen in Figures 3B, 6A, C, 7A-C. No better demonstration could be possible.

2) Figure 2F shows another opening with two "seeds" exposed in the edge in the top area, and this opening seems to be natural too.

It is only a result of natural breaking.

Could it suggest the plant of gymnosperms?

No. The original status of the “seeds” cannot be visualized when it is not broken or altered. The original status can be seen in Figures 4C, 5H. Figure 2F cannot stand for the original status, just as broken pieces of a glass cannot represent an intact glass.

Authors really need to demonstrate the nature of this opening.

We have addressed this concern in the newer version. Breaking is not an opening.

Also interestingly, the two exposed "seeds" in Figure 6H exhibit different colored zones. Are they real seeds or leaf modifications with veins, such as bracteoles?

We have never seen any bracteoles like these *and* in this position. If you still insisted on this, we wish you could share related reference.

Figure 6G shows a seed with two horizontal groves. How to interpret the "groves"?

We cannot confirm the so-called groves in this case and we have no plausible explanation for this. This is the only case in over hundred specimens. More consistent occurrences will make robust interpretation possible and necessary.

Suggestion: apply fluorescent microscopy to examine the "seeds", to see if new evidence can be found.

Thanks. We have performed fluorescent microscopic observation and have found no meaningful information, either of pollen grains or of seeds.

3) The most bizarre structure is the "dendroid style". How many "flowers" have it? If it is a style, the end of each branch should be a stigma. Even if pollen germinated pollen tubes, the pollen wall could still be attached onto the stigmas. Suggestion: use fluorescence microscope and SEM to scan over those branch ends to make sure they are real fossil stigma with pollen.

Thanks. We have performed fluorescent microscopic and SEM observations and have seen no trace of pollen grains.

4) The "stamen" presented in Figure 7J-K is not so convincing so that even the authors call it "possible stamen". Suggestion: make serial sections of specimens of flower buds to figure out the real shape of the stamen and find out more *in situ* pollen. It will be very significant if the pollen type matches up with those found on the "stigma".

We have downplayed the information of stamen, due to singular observation and lack of significant information. We prefer to leave this for future study.

5) The "bracts", "scales", "sepals", and "petals" appear all similar (Figure 5I), including cellular structures (Figure 10), except for size. Even the authors use "b" for a scale in Figure 2I. Could they be all bracts and bracteoles like those on the cones of gnetophytes? How to distinguish them from each other?

It is expected that these from the same plant share a certain similarity in cuticle with each other. We have given detailed explanation in the newer version about the difference among these four foliar parts. We have no idea how this is related to those in Gnetalean cones (of Ephedra, Welwitschia, or Gnetum?).

Reviewer #3:

The principal goal of this manuscript is to describe and present a new fossil taxon *Nanjinganthusdendrostyla*, an angiosperm from Early Jurassic South Xiangshan Fm. that outcrops in China. This new proposed taxon is based on "284 individual flowers on 28 slabs preserved in various orientations and states". They specimens are impressions/compressions. I have major concerns about the interpretation of these fossils and the inclusions of such interpretations within the diagnosis.Unfortunately, I have to disagree with the opinion of the authors that the fossils are well-preserved; the fossils are indeed poorly preserved (the authors even need to add lines on the fossil photos to show "organs" and characters). They placed this proposed taxon within the flowering plants based on their interpretation of structures that the authors claim they are observing. This is clear in the diagnosis (which it is basically a description and not a diagnosis).

Thanks. We assume you are saying that our description is consistent with diagnosis.

The diagnosis, description, treatment, Discussion and conclusions are not consistent with the evidence presented in the figures.

Which part is not consistent? Please be specific so we can correct these mistakes.

These fossils can easily be conifer shoots preserved in different positions (the comparisons and Table 1 presented in the subsection “Eliminating Alternative Interpretations” are extremely vague) for example the cuticular characters of the "petals" are typical of conifers.

The cuticular characters in Figure 9 show elongated epidermal cells in files, and simple pits that we interpret to be isolated stomatal complexes. Far from being “typical of conifers” this combinations of features is found across a very wide range of plant groups, from various gymnosperms and angiosperms. There is nothing about them which would suggest conifer above any other group. If the reviewer insisted on this point, we would love to see the reviewer sharing reference about this and we will address this point specifically.

[Editors' note: the author responses to the re-review follow.]

After discussion between the reviewers we do have two more important suggestions. We decided it would be good to keep the description of the unattached potential stamen, but we strongly suggest that in the text and the legend for Figure 3GH it explicitly explains the relationship of the organ and the associated flower (that they are not attached).

Yes, we agree with this point and have deleted the related information throughout.

In addition, the holotype is not a rock block but an individual fossil. You need to pick one fossil with sufficient detail as the type specimen.Yes, we have designated the holotype and isotypes.

Reviewer #1:

Need corrections:Zeng et al., 2014 is not referenced.

This now is among the references.

Holotype. The holotype is not a rock block but of an individual fossil. You need to pick one fossil with sufficient detail as the type specimen.

Yes, this suggestion has been followed.

Reviewer #2:

I am very pleased to find that this version of the manuscript is in a much better level now, and I would recommend it for publication, with the following aspects to be changed or addressed:1) Insert "Noncarpellate Epigynous" between "Unexpected" and "Flower" in the title, because noncarpellate flower is more significant than its occurrence in Jurassic in my opinion, and epigynous flower is also very important. So, the "Unexpected" refers to the three aspects, and will attract more attention from not only paleobotanists but also modern botanists.

Yes, this suggestion has been followed.

2) Therefore, "Noncarpellate Epigynous" should be emphasized more in the Discussion! For the part "This discrepancy suggests either.…. reassessed", I would recommend to adopt the style of Sattler and Lacrois's statement, "With regard to the evolution of basal placentation in Basella and other taxa of Angiosperms three possibilities exist: 1) It is derived from the carpellate condition, 2) It is primitive and the carpellate condition is derived, 3) Both carpellate and noncarpellate organizations have coexisted during the evolution of Angiosperms which may have been monophyletic or polyphyletic". And *Nanjinganthus* supports their second possibility! The three aspects and their implications should also be mentioned again in the Conclusion.

Yes, this suggestion has been followed.

3) Since the "possible stamen" is not connected to any flower, please remove any related part thoroughly, even from "remarks", Figures, and the reconstruction drawing. Introducing this un-connected "possible stamen" will only weaken the credibility of whole paper!

Yes, this suggestion has been followed.

[Editors' note: further revisions were requested prior to acceptance, as described below.]

The manuscript has been improved but there are some remaining issues that need to be addressed before acceptance, as outlined below:1) That the authors temper their conclusion that their data demonstrates that this is an early Angiosperm flower.2) That the authors make sure the U-Pb analyses are explained in Materials and methods, and the results are properly presented.3) An important source for confirming the age comes from the pollen assemblage. But there is no reference to a comparison from the literature. That the authors provide references and a brief discussion how your assemblage is similar, or a temporal range chart of the palynomorphs and a brief discussion why it supports the age.

We have adopted almost all the suggested changes except a couple places and we have added dating method information and references. For the Santos et al. reference, we provided DOI number. The Figure 11 in the preceding version was short of a line for 5. In the newer version, this line is added, therefore I have also replaced Figure 11. We hope this version can be acceptable now.